# Small-Molecule RAS Inhibitors as Anticancer Agents: Discovery, Development, and Mechanistic Studies

**DOI:** 10.3390/ijms23073706

**Published:** 2022-03-28

**Authors:** Shaila A. Shetu, Debasish Bandyopadhyay

**Affiliations:** 1Department of Chemistry, The University of Texas Rio Grande Valley, Edinburg, TX 78539, USA; shailaakter.shetu01@utrgv.edu; 2School of Earth Environment & Marine Sciences (SEEMS), The University of Texas Rio Grande Valley, Edinburg, TX 78539, USA

**Keywords:** pancreatic cancer, lung cancer, colon cancer, small-molecule inhibitors, heterocycles, KRAS, HRAS, NRAS, RAS subfamily, mechanism of action, cell signaling

## Abstract

Mutations of *RAS* oncogenes are responsible for about 30% of all human cancer types, including pancreatic, lung, and colorectal cancers. While *KRAS1* is a pseudogene, mutation of *KRAS2* (commonly known as *KRAS* oncogene) is directly or indirectly associated with human cancers. Among the *RAS* family, *KRAS* is the most abundant oncogene related to uncontrolled cellular proliferation to generate solid tumors in many types of cancer such as pancreatic carcinoma (over 80%), colon carcinoma (40–50%), lung carcinoma (30–50%), and other types of cancer. Once described as ‘undruggable’, RAS proteins have become ‘druggable’, at least to a certain extent, due to the continuous efforts made during the past four decades. In this account, we discuss the chemistry and biology (wherever available) of the small-molecule inhibitors (synthetic, semi-synthetic, and natural) of KRAS proteins that were published in the past decades. Commercial drugs, as well as investigational molecules from preliminary stages to clinical trials, are categorized and discussed in this study. In summary, this study presents an in-depth discussion of RAS proteins, classifies the RAS superfamily, and describes the molecular mechanism of small-molecule RAS inhibitors

## 1. Introduction

The human body is one of the most complicated biological systems, where hundreds to thousands of biochemical, biophysical, and physicochemical transformations occur at every moment. Most of these biotransformations are catalyzed by enzymes or associated with enzymes/proteins. Several studies found that a set of genes responsible for cancer pathogenesis originated from the transforming activity of the Harvey and Kirsten murine sarcoma retroviruses and referred to them as *RAS* (rat sarcoma virus) in the early 1980s [1]. The initial study of *RAS* identified them as viral genes. They were then attributed to the rodent genomes and thought to be the cause of the oncogenic properties of tumor viruses [2]. Later, in 1982, the first human RAS genes and RAS proteins were identified, and their mutational activation was observed in human cancer cell lines. Three of the four deadliest cancers in the US, lung, colon, and pancreatic cancer, show *RAS* mutations’ prevalence, which also extends to other human cancers [3].

The human RAS protein family now includes more than 150 proteins encoded by genes: *HRAS*, *KRAS*, and *NRAS*. It is known that 86% of mutations in the RAS protein, which constitutes mutated RAS proteins in cancer, come from the one most closely related to the *KRAS* genes. *KRAS* genes are responsible for several types of cancer, and 90% of pancreatic cancer cases are directly attributable to *KRAS* mutations [2,4]. Along with the specific types of cancers, RAS proteins are also responsible for other diseases such as RASopathy, Capillary Malformations, and Psychiatric and Neurodevelopmental Disorders [5].

Studies found that aberrant *RAS* mutation is associated with hyperproliferation developmental disorder and specific mutations in codons 12, 13, and 61. GTP-binding occurs within the particular site and activates RAS signaling. Importantly, all RAS isoforms share sequence identity in all the regions responsible for GDP/GTP binding, GTPase activity, and effector interactions, which suggests functional redundancy. Several synthetic and natural small molecules were evaluated in RAS-dependent cell lines, and mechanistic investigations were carried out in many instances. Despite unceasing progress, many miles remain in the development of drugs with higher potency and lower toxicity.

## 2. Classification of RAS Protein Superfamily

The RAS protein superfamily comprises small guanosine triphosphatase chemicals (GTPases) that act as molecular on/off switches for numerous cellular activities in response to extracellular signals. The RAS family of proteins functions as a set of signaling nodes triggered by various extracellular stimuli. They also control intracellular signaling. Such signaling eventually regulates gene transcription and influences fundamental processes such as cell growth and differentiation [6,7,8,9,10]. Based on their genetic sequences and biological activities, we found five distinct groups of the RAS superfamily: RAS, Rho, Arf/Sar, Ran, and Rab (Figure 1) [6,8].

### 2.1. RAS Subfamily

There are 36 proteins, collectively called RAS oncoproteins, that are the original members of the RAS family. This subfamily has been the topic of much research. The RAS family of human oncogenic members has been intensively investigated, and they modulate cell proliferation, differentiation, morphology, and apoptosis in general. In 1993, the protein kinase Raf was studied and recognized as a RAS effector. RAS is activated when it binds to the Raf serine/threonine kinase. Activated RAS facilitates translocation to the plasma membrane, where additional phosphorylation events enhance Raf kinase activation in its entirety [1,9]. The RAS proteins HRAS, KRAS, and NRAS (H, K, NRAS) are RAS oncoproteins that cause human cancers. The transfusion-based assay can isolate them [10,12,13] from human tumor cells. These three isoforms are most prevalent in human cancers. In addition, the same proteins have recently been linked to central nervous system neuronal plasticity. Among the signaling molecules that are indirectly linked to various cell surface receptors, RAS proteins play a vital role in responding to a wide range of extracellular observations [14].

Most RAS subfamily proteins are primarily found in the plasma membrane area. Therefore, C-terminal prenylation contributes to membrane localization. There is a common Caax (a = aliphatic, x = terminal amino acid) pattern that most prenylation signals follow. This pattern guides cysteine farnesylation (except when x = L or F, which leads to the geranylgeranylation process, commonly seen on RRAS and some other RAP proteins). During the prenylation reaction, proteolysis of the three C-terminal residues (aax) occurs, and methylation of the lipid-modified cysteine is completed [15,16]. RAS subfamily proteins such as the Rap, RRAS, Ral, and Rheb proteins help control the signaling network. RRAS participate in various complex activities such as angiogenesis, regeneration, vascular homeostasis, cell adhesion, and neuronal axon guidance, but their mutation causes invasive cancers [17,18]. Mitogenic stimuli activate Rap proteins, which operate as the controllers of integrin-mediated cell adhesion as well as cell spreading [19]. Ral proteins’ functions include mitogenic responses, protein trafficking, differentiation, and cytoskeleton dynamics. Investigations found that the two Ral proteins play independent and complementary roles in cell transformation [20]. RalA promotes anchorage-independent cell proliferation, but RalB is required to survive against tumor cells [21]. The Rheb protein is largely involved in regulating the cell cycle [22]. Some early studies found that the Rheb protein also performs two additional functions: blocking of MAPK or mitogen-activated protein kinase signaling; inhibiting the RAS-mediated transformation of cultured fibroblasts [23].

### 2.2. Rho Subfamily

The RAS homolog (Rho) subfamily is closely related to the RAS subgroup. The Rho family mainly operates in signaling networks that subsequently regulate three processes: actin, cell cycle progression, and gene expression [24]. In addition, Rho GTPases are critical cytoskeletal regulators that influence various cellular activities, including cell polarity, migration, vesicle trafficking, and cytokinesis. Rho GTPases have been investigated extensively in several mammalian cell types, with dominant-negative and constitutively active mutants primarily being employed [25]. Rho GTPases have also been identified as crucial regulators of cell polarity. They act as molecular regulators between two states: active GTP–bound and inactive GDP–bound [26]. In addition, many studies have implicated members of the Rho family in hematopoiesis. Multiple human hematologic disorders, including neutrophil dysfunction, leukemia, and Fanconi anemia, have been associated with the deregulation of Rho GTPase family members, increasing the possibility that Rho GTPases and their downstream signaling pathways could be helpful therapeutically [27]. RAC proteins have emerged as essential mediators of Wnt signals, mainly in noncanonical and canonical pathways leading to β-catenin-dependent transcription. Wnt-induced signals appear to promote morphological and transcriptional alterations that affect cell behavior spatially and temporally [28].

To understand the Rho family’s evolutionary history, there have been investigations on over 20 species, including major eukaryotic clades, unicellular organisms, various mammals, platypus, opossum, and so on. Studies include the reconstruction of the ontogeny and chronology of the development of distinct Rho subfamilies. There are eight subfamilies among the 20 mammalian Rho members. The Rac, Rho, and Cdc42 subgroups have been extensively studied among the eight Rho subfamilies [29]. Most likely, 20 canonical Rho members (Rho GTPases) are regulated by four guanine nucleotide dissociation inhibitors (GDIs), 85 guanine nucleotide exchange factors (GEFs), and 66 GTPase-activating proteins (GAPs) [30].

### 2.3. Rab Subfamily

Rab proteins are small GTP-binding proteins, first characterized as RAS-like genes exhibited in rat brains [31], that comprise the biggest subfamily of the RAS protein superfamily. The yeast *Saccharomyces cerevisiae* expresses eleven Rab (Yptp/Sec4p) proteins. However, based on the expressed-sequence tags (ESTs) and the sequenced human genome, there could be as many as 63 Rab (Yptp/Sec4p) proteins in humans. Rab proteins act as membrane-associated molecular switches between different organelles to regulate vesicular trafficking pathways. The Rab subfamily promotes vesicular or tubular carriers’ budding in further steps such as the donor-compartment, transport-to-acceptor, vesicle-fusion, and load-release steps [32,33].

Rab GTPases are connected in reverse to membranes by hydrophobic geranylgeranyl groups attached to one or (usually) two carboxy-terminal Cys residues, which is essential for controlling the membrane traffic. Rab proteins are regulated by their ability to act as molecular switches, similar to the other GTPases, that wiggle between GTP- and GDP-bound conformations.

Each transport step starts with the binding of activated Rab proteins to soluble factors that function as ‘effectors’, allowing the transduction of the Rab GTPase signal. Several RAB effectors have been identified. Rab effectors’ structural heterogeneity suggests a collection of highly specialized molecules that function exclusively on specific organelles and transport systems [34]. Rab pathway dysfunction has been linked to diseases such as immunodeficiencies, cancer, and neurological disorders.

### 2.4. Ran Subfamily

Ran is a member of the RAS superfamily, which is the most abundant small GTP-binding protein. Ran proteins are localized mainly within the nucleus [35]. These proteins regulate the nuclear import/export of materials. In addition, DNA replication, mitotic spindle assembly, and nuclear envelope assembly [36] are also controlled by these proteins. Although the RAN protein is related to the Rab subfamily, it has characteristics that set it apart. Ran function depends on a spatially bounded form of the GTP of Ran, unlike other small GTPases. Nuclear trafficking studies identified the switching behavior of Ran. It was found that Ran functions as a molecular switch and controls nuclear transport receptor–cargo complexes [37,38]. The assembly and disassembly of nuclear transport receptors depend on the guanine-nucleotide-bound state of Ran.

### 2.5. Arf Subfamily

Similar to the Rab protein family, the Arf family of proteins is another diverse group of proteins that are involved in vesicle trafficking [6]. This subfamily’s first members function as ADP ribosylation factors, and they were named accordingly. Arf proteins play a role in vesicle formation and structure, as well as intracellular vesicular transport, endocytosis, and exocytosis [39,40,41]. The Arf family consists of 30 genes, divided into four groups:The ArfsThe Arf-like (Arls) proteinsThe Secretion-associated RAS-related (SARs) proteinsThe single Tripartite protein 23, also known as TRIM23.

ARF GTPases constitute a significant family of eukaryote-specific GTPases. They regulate membrane traffic, actin dynamics, tubulin assembly, and cilia-related functions. Every Arf isoform has several biochemical effects that can be put together to elicit a specific cellular behavior or event. This phenomenon helps researchers assess the pathway’s potential as a target in disorders such as cancer [42,43].

### 2.6. KRAS, HRAS, and NRAS

Three RAS proteins, HRAS, KRAS, and NRAS, are expressed in all mammal cells and, thus, are closely related. Hence, the aberrant mutation of these three proteins can contribute to the development of oncogenesis, which is often attributed to a unique mutation, and this typically occurs at codons 12, 13, or 61. Investigations have found RAS mutations in 30% of human cancers. Though isoforms have a lot in common, each isoform prefers to couple to specific cancer types. Therefore, there has been much analysis of the mutational spectra of RAS isoforms. One such study shows that factors other than the differences in mutagen exposure exist that contribute to the heterogeneity observed across a single isoform.

#### 2.6.1. KRAS

The *KRAS* gene is found to be mutated in various types of tumors, such as pancreatic carcinoma (over 80%), colon carcinoma (40–50%), lung carcinoma (30–50%), and others [11,43,44,45,46]. The KRAS isoform causes about 85% of RAS mutations. Human *KRAS* oncogenes are activated in two ways: when oncogene copies increase abruptly or when the tumor suppressor gene p53 is lost abnormally. When oncogenic *KRAS* is activated, a mutant *KRAS* protein remains actively GTP-bound for an extended period. KRAS cycles between two states: an inactive bound GDP state, and an active GTP-bound state. However, only the GTP-bound state of KRAS is necessary for Raf-kinases, PI3K, and RalGDS to bind and consequently activate their effector proteins [47].

#### 2.6.2. HRAS

HRAS mutation comprises 12% of overall RAS mutations. HRAS is one of three human RAS proteins that encode the GTPase HRAS protein. HRAS mutations are found in bladder urothelial carcinoma, breast-invasive ductal carcinoma, lung adenocarcinoma, prostate adenocarcinoma, and colon adenocarcinoma, with these having the most significant prevalence of alterations. However, HRAS mutations are also prevalent in the bladder (4.5%), thyroid (3.2%), and head-and-neck carcinoma (5.1%) [48].

#### 2.6.3. NRAS

NRAS is the minor cancer-causing mutant among the three human RAS proteins, and approximately 2.3% of overall RAS mutations occur because of the NRAS mutant. Oncogenic mutations of NRAS genes in human tumors follow a different distribution pattern, with the highest rates of mutation found at codon Q61 (about 60% of total NRAS mutations) and lower percentages detected at G12 (24.4%) and G13 (12.7%) [49]. Moreover, it was found that NRAS mutations are prominent in hematopoietic and skin (melanocyte) malignancies and myelomas [1].

## 3. RAS and Cancers

### 3.1. Pancreatic Cancer

Most cases of pancreatic cancer have mutations in the *KRAS* gene at the time of diagnosis (>80% of cases). It was found that 90% of pancreatic cancers are developed in the exocrine compartment of the pancreas and cause pancreatic ductal adenocarcinoma (PDAC). Codon 12 is accountable for 98% of PDAC mutations. Research found that codon G12D (51%) is the most aggressive PDAC subtype, followed by G12V (30%), G12A/C/S (2% each), and G12L/F (<1%) (Figure 2) [50]. *KRAS* plays a vital role in developing PDAC-type pancreatic cancer [51]. The progression of PDAC occurs due to the loss of the tumor-suppressor gene cyclin-dependent kinase inhibitor 2A (*CDKN2A*) [52]. Although *KRAS* mutations are undruggable and no anti-*KRAS* therapies exist for them, some inhibitors show effects against cancers. Inhibition of the RAF-MEK-ERK protein kinase pathway is one of the most potent factors that work against *KRAS* mutation. This pathway has an important role in the progression of PDAC [53]. It is also reported that iExosomes inhibit PDAC in mice by delivering RNAi. However, exosomes have therapeutic potential to control *KRAS*-dependent pancreatic cancer [54,55]. Additionally, the *KRAS* mutation can be inhibited by using a slow-released biodegradable polymer matrix. This process works by delivering siRNA as an extended-release drug to mutated *KRAS* [56]. Another method for suppressing the *KRAS* mutation is anti-RAS vaccination. These vaccines contain different mutated genes that work against *KRAS* mutations [57,58].

### 3.2. Lung Cancer

Lung cancer ranks as the primary cause of cancer-related death among men and women worldwide. Researchers first identified activation of *KRAS* mutations to cause lung cancers. *KRAS* mutations are responsible for the majority of lung adenocarcinoma (LADC) and are also seen in non-small-cell lung cancer (NSCLC). *KRAS* mutations cause 30% of lung carcinoma cases. Significantly, 90% of RAS mutations in LDAC are *KRAS* mutations, and this figure is around 97% in NSCLC [61,62]. Furthermore, the study suggests that *KRAS* mutations are more frequent in African American patients than in Asian patients (Figure 2) [63,64]. An investigation of restricted NSCLC tumors, in particular, indicated that *KRAS* mutations are more common in younger patients, especially young women [65].

Researchers determined that smoking history is essential in the regulation of *KRAS* mutation in lung cancer. Studies found that a significant amount of lung cancer patients are cigarette smokers, and *KRAS* mutation is also frequent in adenocarcinomas. Furthermore, it was also identified that 30% of current and former smokers exhibit lung adenocarcinoma. However, *KRAS* mutation is also found in never-smokers, but the percentage of *KRAS* mutations is higher in smokers [50,66].

### 3.3. Colorectal Cancer

Colorectal cancer (CRCs) is one of the principal causes of death in men and women in the United States [67]. Multiple genetic alterations make colon carcinoma more fatal. About 50% of *KRAS* mutations are responsible for codon 13,14, and 64 in colon adenomas [68,69]. Nearly 40% of *KRAS* mutation occurs for G12D, G12V, and G13V that activate the MAPK signaling pathway (Figure 2) [70]. Although extensive studies have been conducted to find the therapeutics in CRC treatment, there is still no direct inhibitor of *KRAS* mutations [71]. More effective treatment and biomarkers are needed because a large percentage of colon cancer is associated with the *KRAS* oncogenes.

### 3.4. Other Cancers

RAS mutations comprise a vast portion of cancer-causing diseases. RAS mutations also occur in other cancers such as breast cancer, head-and-neck cancer, bladder tumors, skin cancer, prostate cancer, etc. More attention is needed to identify the RAS inhibitors. 

## 4. Small-Molecule RAS Inhibitors

Small molecules are drugs that impede the cellular pathway or molecules within the pathway. Small-molecule drugs/inhibitors can enter the cell easily because of their low molecular weight and balanced ADME properties. They can target the extracellular cell surface receptor as well as intracellular proteins. After entering the cells, they inhibit other molecules such as proteins or enzymes, which causes various types of disease. For example, gefitinib is a small-molecule anticancer drug used to treat non-small-cell lung cancer [72]. Gefitinib blocks the signaling pathway of epidermal-growth-factor receptors (EGRF), which are responsible for cell proliferation, invasion, etc. RAS proteins contribute considerably to many types of cancer. There is no more effective treatment for *RAS* mutations. In terms of finding an effective means of treating undruggable *RAS* mutation, small-molecule drugs or inhibitors can be an effective option for cancer treatment, inspired by robust investigation and research.

### 4.1. Classification of Small-Molecule RAS Inhibitors Based on Their Structure

Nowadays, structure-based drug discovery is becoming an essential strategy for treating various diseases, especially cancer. This strategy is now in vogue because of its cost-effectiveness and faster results compared to traditional drug-discovery methods. However, structural data provide the molecular mechanisms and fundamental functions of a compound. Structural studies have provided hundreds of new targets, and when the target is identified, it is essential to obtain accurate structural information.

This study classified and discussed small-molecule RAS inhibitors according to their structure. We found that most of the RAS inhibitors are heterocyclic compounds. The cyclic part (from the Greek “kyklos”, meaning “circle”) of “heterocycle” indicates that at least one ring structure is present in such a compound, while the prefix hetero- (from the Greek “heteros”, meaning “other” or “different”) refers to any atoms other than carbon (heteroatoms) [73]. The most common heterocycles have four- or five-membered rings containing nitrogen, oxygen, and a sulfur atom. These diverse heterocyclic compounds have found applications as pharmacophores. These pharmacophores are often found in synthetic and natural product drugs. Researchers and pharmaceutical companies are interested in heterocyclic pharmacophores due to their fascinating medicinal properties. We also found that the carbocyclic compound also has RAS-inhibitory properties. Thus, studying the heterocyclic and carbocyclic RAS inhibitors might be an excellent resource for finding potent small-molecule inhibitors for cancer treatment.

Based on the structure aza heterocyclic small-molecule RAS inhibitors with one nitrogen atom are shown in Table 1. After careful review, the mechanistic properties, cancer type and major pharmacophores are also described.

#### 4.1.1. Ganetespib

The heat-shock protein 90 (Hsp90) is inhibited by ganetespib (Table 1) leading to the degradation of many oncogenic proteins such as v-Raf murine sarcoma viral oncogene homolog B, VEGF receptor, EGFR, Cyclin D1, and mutant p53 [74]. Acquaviva et al. [75] found that ganetespib, a small-molecule inhibitor, suppresses human lung cancer in individualized therapy. Ganetespib showed a potent cytotoxic effect in both in vitro and in vivo studies by inhibiting Hsp90. A set of twenty NSCLC cell lines with known *KRAS* mutations, including the G12, G13, and Q61, variants, were targeted to identify the cytotoxic activity of ganetespib. Cell lines H727 and H441 showed IC_50_ values of 28 and 14 nmol/L against the *KRAS*^G12V^ mutant, respectively, which indicates that the small-molecule inhibitor ganetespib is responsive to *KRAS* mutation. Additionally, ganetespib and the PI3K/mTOR inhibitor BEZ235 showed potential effects in a A549 xenograft model. Finally, a phase II clinical study specified 47% tumor shrinkage in the *KRAS*-harbored mutation. In 2017, Cercek et al. [74] hypothesized that ganetespib might be an excellent small-molecule inhibitor in colorectal cancer as *KRAS*^G12V^ is one of the main mutants responsible for apoptosis, cell cycle control, and angiogenesis [81,82]. However, in a phase II study with seventeen patients, significant antitumor activity was not shown with Ganetespib (200 mg/m^2^) treatment when administered intravenously. Interestingly, out of nine patients (3 G12D, 4 G12V, 1 G12S, and 1 G12C), two patients, harboring G12V mutations showed stable conditions.

#### 4.1.2. Apatinib

Apatinib (Table 1) is a small-molecule tyrosine kinase inhibitor that significantly works against metastatic breast cancer and gastric cancer by selectively inhibiting vascular endothelial growth factor receptor-2 (VEGFR-2) [83,84]. In 2017, it was reported that the dosing effect might vary the effectiveness of apatinib in different types of cancer [76]. Their results for four advanced-stage (stage-IV) patients with *KRAS* mutation, aged 56–81 years, who were diagnosed with metastatic lung adenocarcinoma indicated the potential positive effect of apatinib. Among the four patients, three showed progression-free survival (PFS) (1.5 months, 4.5 months, and 5.5 months) with a small dose. Only one patient was diagnosed with manageable side effects (hoarseness and hemoptysis). Their implemented dosing amount (250 mg/d oral) was significantly lower than the previously studied large-scale trial (500–850 mg/d), in which significant side effects were observed.

#### 4.1.3. Oncrasin-1

A small-molecule compound, oncarsin-1 (Table 1) was identified by synthetic lethality screening and was found to significantly suppress various lung cancers and might inhibit the novel *KRAS*/PKCι pathway [77]. Although oncarcin-1 has a similar core structure to indole-3-carbinol and lonidamine, an investigation found that it might follow different anticancer mechanisms. This is because indole-3-carbinol [85] and lonidamine [85] have no cytotoxic effect on T29, T29Kt1, T29Ht1, and H460 lung cancer cell lines, whereas oncarcin-1 shows anticancer effects in these cell lines. When T-29 (or T29Kt1) cells were treated with 10 μmol/L oncarsin-1 or H460 cells were treated with 1 μmol/L of oncarsin-1 (equivalent to approximately IC_80_ doses) the apoptotic cell counts were 33.2%, and 47.2% respectively, which indicated that oncarsin-1 has an effective apoptosis-induction mechanism. Additionally, to identify the *KRAS*-mutated cell death in H460 cells, orcarsin-1 treatment with control siRNA-treated cells and *KRAS* siRNA-treated cells showed approximately 17% and 3% more apoptosis, respectively, when compared to treatment with DMSO + siRNA-treated cells. In an in vivo study, oncarsin-1 treatment was compared to solvent-only treatment, and the result showed 75.4% more tumor suppression with oncarsin-1 therapy [77].

#### 4.1.4. *N*-(1-Acryloylazetidin-3-yl)-2-(5-bromo-3-(5-methoxy-1,2,3,4-tetrahydroisoquinoline-2-carbonyl)-1H-indol-1-yl) Acetamide

Shin et al. found a series of novel *KRAS*^G12C^ inhibitors by exploring the chemotype evolution (custom library synthesis accompanied by subsequent screening) and structure-based design strategies [78]. The newly synthesized compound (†) (Table 1) showed the most potent and selective inhibition against *KRAS*^G12C^ mutants. The path of optimization of this series of small molecules identifies the hidden surface groove bordered by the side chain of Y96/H95/Q99 amino acids. Mechanistically, *N*-(1-acryloylazetidin-3-yl)-2-(5-bromo-3-(5-methoxy-1,2,3,4-tetrahydroisoquinoline-2-carbonyl)-1H-indol-1-yl) acetamide improves covalent interaction with specifically targeted proteins by means of submolecular inhibition [78]. The first covalent irreversible interaction of *KRAS*^G12C^ was identified by application of tethering, and allosteric binding of the compound in the P2 pocket (near switch II region) resulted in the inhibition of RAS mutation [86]. Further study found more successful results using electrophile screening. An irreversible interaction was produced by means of chemotype evaluation of the anchor molecule (bait), which helped to inhibit the *KRAS*^G12C^ mutation. Shin et al. reported an *N*-(1-acryloylazetidin-3-yl)-2-(1*H*-indol-1-yl) acetamide that modifies *KRAS*^G12C^ by most actively hitting with the indole bait and covalently binding with the switch II region. After checking the cytotoxic effect on MIA PaCa-2 cell lines, the compound demonstrated an IC_50_ value of 0.638 μM. Further investigation proved that adding methyl group in two positions of the indole group increased the activity two-fold (IC_50_ = 0.299 μM) and two-fold (IC_50_ = 1.68 μM) in the GTP 2 h exchange and cellular potency assays, respectively, in comparison to the previous compound [78]. A recent report showed that Y96/H95/Q99 are mutated in cancer cells under the influence of *KRAS*^G12C^ inhibitors [87].

#### 4.1.5. 2-((4-((1-(2-(2,4-Dichlorophenoxy) acetyl) piperidin-4-yl) amino)-4-oxobutyl) disulfaneyl)-N,N-dimethylethan-1-aminium

A newly developed small molecule (Table 1) exerts a therapeutic effect by inhibiting *KRAS*^G12C^ mutants and binding to the allosteric pocket. This small-molecule inhibitor is attached to the Cys of *KRAS*. It binds the allosteric binding site very close to the switching II pocket (SW IIP), resulting in the disruption of GTP-state RAS conformation and the impairment of Raf activity [79]. Although it is difficult to identify the specifically targeted allosteric binding site for a specific protein, cystine-dependent small molecules can irreversibly bind with the allosteric binding site of the *KRAS*^G12C^ oncoprotein.

#### 4.1.6. GDC-0449 (Vismodegib)

The combination therapy of a small-molecule hedgehog (Hh) inhibitor (GDC-0449) (Table 1) (small hydrophobic molecule) and miRNA (miR-let7b) (oligonucleotide) can inhibit pancreatic cancer growth both in vitro and in vivo [80]. Researchers found that increasing the level of Hh signaling promotes cell proliferation by controlling EMT and PI3 in a kinase-dependent manner, which results in different types of cancer (pancreatic cancer, breast cancer, colon cancer, prostate cancer, brain tumor, and basal cell carcinomas). Hh signaling is also responsible for CSCs proliferation and decreases apoptosis by regulating Bcl-2 and Bcl-X. However, GDC-0449 in monotherapy is efficient for tumor cell proliferation in vitro via the inhibiting of Hh signaling [88]. While, due to the lack of efficient delivery, miR-let7b cannot exert an effect on PDAC, it was reported to cause pancreatic cancer cell growth inhibition [89]. Therefore, cationic chains of mPEG-b-PCC-g-DC-g-TEPA copolymeric micelles are used as a suitable delivery system for miR-let7b. Finally, mPEG-b-PCC-g-DC-g-TEPA copolymeric micelles encapsulate GDC-0449 and form a complex with miR-let7b. The synergistic effect of these two inhibitors combined inhibited pancreatic cancer in an in vitro and in vivo study [80].

### 4.2. Aza Heterocyclic Small-Molecule RAS Inhibitors with More Than One Nitrogen Atom

In Table 2 Aza heterocyclic small-molecule RAS inhibitors with more than one nitrogen atom are classified. 

#### 4.2.1. ARS-1620

The current study was successful in considering *KRAS* as a therapeutic target. The new investigation might be a promising therapeutic approach for *KRAS*-dependent cancer treatment. Article [90], which explored the *KRAS*^G12C^ S-IIP-binding site in a structure-based design study, showed KRAS inhibition through the invention of a new covalent compound (ARS-1620). ARS-1620 (Table 2) is the first orally bioavailable, potent, and selective small-molecule *KRAS*^G12C^ inhibitor. Moreover, it also provides evidence of therapeutic inhibition for both in vitro and in vivo studies. ARS-1620 treatment showed drastically different results with 2D monolayer adherent and 3D ultra-low adherent suspension in in vitro and in vivo studies [90]. Three-dimensional ULA spheroid culture showed a much more effective result than 2D monolayer culture. This result suggested an inherent difference in *KRAS* dependence in different types of culture cells.

#### 4.2.2. ARS-853

Further studies were conducted to increase the potency of ARS 1620. An investigation found that this small-molecule compound did not show any cellular response when bound with non *KRAS*^G12C^, even with 10-fold higher concentrations, but targeted the GDP-bound protein ARS-853 (Table 2) and inactivated the signaling of *KRAS*^G12C^ with an IC_50_ value of 1 μmol/L, indicating the selectivity of ARS-853 [91]. As a result, a more potent and selective covalently bounded *KRAS*^G12C^ inhibitor was identified.

#### 4.2.3. AMG 510 (Lumakras or Sotorasib)

To make the ARS-1620 more efficient and draggable, Canon et al. [92] developed a new small-molecule inhibitor named AMG 510. This compound is superior to ARS-1620, and they have minimal structural differences. The use of an irreversible strategy of the His95 groove, which is close to the cystine pocket AMG 510 (Table 2), produces robust inhibition of the *KRAS*^G12C^ oncogene. It also downregulates the MAPK signaling pathway in both pancreatic and lung cancer. However, it does not affect wild-type *KRAS*^G12C^ mutations, which proves the specificity of AMG 510. After the consequent success, a further study was conducted to obtain a more efficacious result. Combinatorial therapy of AMG 510 with a MAPK inhibitor (carboplatin) created a more potent outcome than monotherapy. These effective outcomes induced researchers to move forward. As a result, synergistic effects were tested in a mice model, and 90% of mice showed complete tumor suppression. Finally, a preliminary clinical study on the human model provided 50% tumor regression. However, most of the success of these comprehensive studies relates only to the *KRAS*^G12C^ oncogene, mostly found in lung cancer [105]. Only 2% of PDAC are responsible for the *KRAS*^G12C^ oncogene [50]. Nevertheless, after reviewing this success, it was suggested that the *KRAS*^G12D^ and *KRAS*^G12V^ mutations, responsible for approximately 80% of pancreatic cancer, might be the primary targets for future studies [50].

#### 4.2.4. MRTX849 (Adagrasib)

Another structurally modified small-molecule inhibitor, MRTX849 (Table 2), developed by Jill Hallin et al. [93], was found to be a very potent, selective, and covalent *KRAS*^G12C^ inhibitor in pancreatic and lung cancer, and is structurally and functionally close to the AMG 510. This compound exerts effectiveness in combination therapy and has remarkable potency when administered alone. Thus, a phase I clinical study was conducted with the single agent in two patients with final-stage lung and colon carcinoma. The results showed partial responses to these two critical types of cancer. However, the return of ERK signaling and the lack of inhibition of mTOR-S6 signaling made the MRTX849 response short-lasting and ineffective. The results from this study influence future expectations for pancreatic cancer treatment.

#### 4.2.5. 1-(2-Hydroxyethyl)-4-(2-Methyl-3,5-Diphenylpyrazolo[1,5-a] Pyrimidin-7-yl) Piperazin-1-ium

Recently, investigations found that AMG 510 is not the only small-molecule inhibitor that is potent in terms of the *KRAS* mutation. In 2019, McCarthy et al. developed a new small-molecule inhibitor [94], which has the affinity to bind to the allosteric binding site. This allosteric pyrazolopyrimidine-based inhibitor (†††) (Table 2) binds with the allosteric p1 pocket of both wild-type and *KRAS* mutants. This new small molecule blocks the MAPK pathway by downregulating the Raf signaling towards RAS mutation. Another advantage of this allosteric inhibition is that it is not limited to only this specific *KRAS*-type allele. It shows benefits for wild-type and GTP-bound subtypes of the *KRAS* oncogene. This remarkable positive result makes this allosteric inhibitor a first-line therapy for tumor treatment in a different range. This small-molecule allosteric inhibitor showed notably good results in lung and oral cancer cell lines that make the allosteric pyrazolopyrimidine-based inhibitor applicable for PDAC treatment.

#### 4.2.6. BGB324 (Bemcentinib)

PDAC progression is directly interlinked with Axl, where the TBK1–NFkB pathway and innate immune suppression are caused by Axl kinase signaling [106,107]. BGB324 (Table 2) is an Axl kinase inhibitor that inhibits colony formation in the PDAC cancer cell line, and gemcitabine produces a moderate therapeutic effect in vitro. BGB324 works by decreasing the phosphorylation of Akt, and TBK1 causes inhibition of Axl signaling [95]. An investigation showed that treatment with BGB324 in six human and three mice PDAC cell lines achieved IC_50_ values ranging from 1 to 4 μmol/L. When BGB324 was administered in combination with gemcitabine, the therapeutic efficacy was identified for a prolonged time. A study on the murine model showed that control drugs or only gemcitabine therapy provide therapeutic effects for not more than one day. Still, applying combination therapy (BGB324+ gemcitabine) exerted a prolonged effect and extends the possibility of novel treatments against PDAC [95].

#### 4.2.7. ABT-737

p21-activated kinase (PAK) is an effector of small GTPase Rac and cdc42 proteins. These proteins are responsible for cell-cycle regulation, cell division, and transformation [108,109]. Studies found that inhibition of PAK4 and PAK1 suppresses the *KRAS* and BRAF mutation in colon cancer in vitro. ABT-737 (Table 2) is a small-molecule inhibitor that represses Bcl-2/Bcl-X_L_, resulting in the inhibition of RNAi of PAK4 and PAK1 in HCT116 colon cancer cells [96]. Although PAK1 and PAK4 phosphorylate are common substrates in phosphorylation pathways (RAF/MEK/ERK) [110,111,112,113], they did not show any consistent inhibition of the RAF1, MEK, or ERK signaling pathways in HCT116 colon cancer cells. The results showed that ABT-737 treatment in PAK4 led to 43–89% reductions in 6 of 7 cells, and the suppression of PAK1 caused 45–50% decreases in 2 of 7 cells according to the cell proliferation assay. Interestingly, when PAK4 and PAK1 were treated simultaneously, only PAK4 responded for seven colon cancer cell lines in the cell proliferation assay. As a result, further investigation identified that PAK4 and PAK1 suppression were responsible for 95% and 80% cell proliferation, respectively, in HCT116 colon cancer cell lines alone [96].

#### 4.2.8. AZD6244 (Selumetinib)

The identification of specific inhibitors for specific biomarkers is an important strategy in cancer treatment. Tentler et al. [97] classified the biomarkers that showed therapeutic responses to the MEK1/2 pathway using the small-molecule inhibitor AZD6244 (Table 2) in *KRAS*-mutated colorectal cancer. They tested AZD6244 in 37 CRC cell lines with *KRAS*/BRAF mutation status. They found IC_50_ values of ≤0.1 mmol/L in AZD6244-sensitive (7 of 27 cells) and of >1 mmol/L in AZD6244-resistant (11 of 27 cells) CRC cell lines. A further study was conducted to classify the genomic target for AZD6244. Tests conducted using in vivo mouse model, and human model found that the *KRAS* mutants had limited therapeutic options and expressed different gene array and pathway analysis [97]. The Wnt signaling pathway was overexpressed in AZD6244-resistant CRC cell lines, indicating that this pathway might be responsible for resistance in the MEK inhibitor and, furthermore, the Wnt pathway might be a potential target for MEK inhibitors via modification of their structure or therapeutic activity [97].

#### 4.2.9. NVP-BEZ235 (Dactolisib)

A small-molecule compound, NVP-BEZ235 (Table 2), was identified as a potent dual pan PI3K/MEK inhibitor in lung cancer. Doxycycline induces p110α H1047R mutation in human lung cancer cells [98]. P110α is a catalytic subunit that is encoded by the PIK3CA gene and activates PI3K [114]. NVP-BEZ235 inhibits p110α H1047R mutation by blocking the kinase activity of PI3K. A further in vivo study was conducted in the mice model to identify the efficacy of NVP-BEZ235 by inhibiting TORC1. However, combination therapy of NVP-BEZ235 and rapamycin successfully decreased the S6 phosphorylation of TORC1, resulting in significant tumor suppression. However, the single treatment of NVP-BEZ235 did not respond in an in vivo study [98]. An investigation of previous research revealed that PI3K caused *KRAS* mutation in the mouse model [115,116]. As a result, different genetic alteration approaches were used in further investigations, and it was confirmed that the downregulation of PI3K suppresses *KRAS*-mutated lung carcinoma, but the effect was not so significant. Recent studies found that the inhibition of both PI3K and ERK signaling can be highly effective in cancer treatment. As a result, the combination treatment of NVP-BEZ235 (PI3K inhibitor) and ARRY-142886 (MEK Inhibitor) [117] showed significant tumor suppression in *KRAS*^G12D^ lung cancer cells, whereas ARRY-142886 alone exerted mild effect [98].

#### 4.2.10. R115777 (Zarnestra or Tipifarnib)

A potent small-molecule inhibitor, R115777 (Table 2), can inhibit farnesyl transferase signaling by inducing apoptosis in myeloma cells. RAS mutation is commonly seen in myeloma and is activated by interleukin receptor 6 [118]. Previous studies found that another FTI inhibited RAS prenylation by activating growth arrest in different myeloma cells, whereas R115777 also showed a similar mechanism in inhibiting myeloma [118]. On the contrary, further investigation showed that R115777 inhibited cell growth following another mechanism that did not involve inhibiting RAS prenylation. These results indicate that R115777 follows an independent RAS mechanism [99]. Apoptosis activity was investigated in Mcl-1, Bcl-XL, and Bcl-2 antiapoptotic proteins [119], which are overexpressed in myeloma [120,121]. It was found that R115777 suppresses Bax overexpression by disputing mitochondrial membrane and activating endoplasmic reticulum stress. To identify the inhibition of caspase-9-induced apoptosis, the activity of R115777 was compared to known caspase-9 inhibitors, and the results showed a dose-dependent inhibition of caspase-9, which indicates that R115777 induces apoptosis by inhibiting caspase-9 [99].

#### 4.2.11. PPIN-1, PPIN-2

Targeting protein–protein interaction (PPI) is an attractive way to treat RAS mutation. Cruz-Migoni et al. [100] identified two PPI inhibitors, PPIN-1 and PPIN-2 (Table 2), which bind to pocket 1 near the effective binding site. However, after binding to *KRAS*_166_^G12D^, these inhibitors cannot impair any *RAS* mutation due to the presence of intracellular anti-RAS antibody fragments. Nevertheless, using a synthetic method, a new compound can be developed by combining the two classes of RAS-binding compounds. As a result, the inhibitor PPIN was combined with one of the RAS-binding intracellular antibody derivatives (Abd-7), and the RAS inducer PPI was inhibited. To bind these RAS inhibitors together, PPIN biphenyl group acted as the bait, and three crossover compounds, Ch-1, Ch-2, and Ch-3, were synthesized. The combination of PPIN-2 and these newly synthesized compounds bound smoothly in pocket 1, and Ch-1 and Ch-3 showed better IC_50_ values (5.3 and 4.5 µM, respectively) than Abd-7 [100].

#### 4.2.12. pan-RAS Inhibitor 3144 (RAS-IN-3144)

Investigations found that it is tough to identify specific inhibitors that bind to the RAS protein in the appropriate binding site [122,123]. Sometimes interacting with allosteric or adjacent binding sites, small-molecule inhibitors can produce therapeutic effects on RAS mutation. Welsch et al. [101] synthesized the pan-RAS inhibitor 3144 (Table 2), which showed a binding effect with the adjacent site of the *KRAS* oncoprotein. After passing different experimental conditions, 3144 was found to bind with *KRAS*^G12D^ and *KRAS*^G12V^. Further investigation found that 3144 induced caspase activity, increased apoptosis, and mechanistically downregulated the overexpression of the RAS effectors PI3K/AKT in addition to RAF/MEK/ERK signaling [101]. Additionally, 3144 inhibited progression and tumor growth in the case of RAS mutation both in vitro and in vivo. A further experiment showed its effectiveness on *KRAS*-mutated pancreatic cancer in a mouse model, but significant side effects were also observed in this case [101].

#### 4.2.13. Deltarasin

Recent studies found that Deltarasin (Table 2) downregulates the RAS/RAF signaling pathway by inhibiting Phosphodiesterase-δ (PDEδ) binding with the hydrophobic pocket of PDEδ, resulting in the inhibition of *KRAS*-harbored pancreatic ductal adenocarcinoma (PDAC) [124,125]. As *KRAS* mutation is responsible for different types of cancers—lung cancer, colorectal cancer, etc.—more investigation is desirable for inhibiting *KRAS* oncogenes. The majority of lung cancer incidents happen for *KRAS* mutants [126]. As a result, Leung et al. [127] first identified that deltarasin induces apoptosis significantly, both in vitro and in vivo, in lung cancer cells. It also induces autophagy in lung cancer cells by inhibiting the MAPK/mTOR signaling pathway. It was also shown that when deltarasin is treated with 3-MA (autophagy inhibitor), it increases autophagic properties and produces more intracellular ROS levels, thus protecting against further autophagy.

#### 4.2.14. (2. R,4aR)-3-Acryloyl-11-chloro-9-fluoro-10-(6-fluoro-2-hydroxycyclohexa-2,4-dien-1-yl)-2,6-dimethyl-2,3,4,4a-tetrahydro-1H-pyrazino [1’,2’:4,5] Pyrazino[2,3-c] Quinolin-5(6H)-one

Kettle et al. [103] identified a potent and selective *KRAS*^G12C^ inhibitor (Table 2) by introducing a key methyl group to the piperazine. The *KRAS*^G12C^ allele constitutes an “Achilles heel” where the small-molecule inhibitors can bind covalently, bind the mutant cysteine, and create an allosteric pocket on GDP-bound RAS. The strategy modifies the weak *KRAS*^G12C^ inhibitor that binds to this specific site, increasing the potency of the newly synthesized compound. The in vivo study showed significant tumor regression in *KRAS*^G12C^ mutant Miapaca2 xenografts.

#### 4.2.15. SML-8-73-1 and SML-10-70-1

Min Lim et al. [104] found that guanine-based small-molecules covalently bind in the guanine-binding site and that binding with covalent inhibitors can irreversibly inhibit *KRAS* signaling. *KRAS* mutations occur near the usual position of the gamma phosphate of the GTP-binding site. After conducting the X-ray crystallography and molecular docking study, a promising candidate (SML-8-73-1) was synthesized, which binds the specific guanine-binding site. To improve the potency, a new analog of SML-8-73-1 (Table 2) was developed by modifying the beta phosphate as an alanine ester phosphoramidite, resulting in SML-10-70-1(Table 2). Moreover, the new derivative, SML-8-73-1, produced antiproliferative effects on the *KRAS*^G12C^-mutated cell line.

### 4.3. Oxoheterocyclic Small-Molecule RAS Inhibitors

Identification of major pharmacophore(s) and biomolecular target based on the structure of oxoheterocyclic RAS inhibitors are shown in Table 3.

#### 4.3.1. NHTD

The binding of *KRAS* with the prenyl-binding protein (PDEδ) produces its oncogenic property. Therefore, inhibition of PDEδ can be an effective path for *KRAS*-induced cancer treatment [125,130]. The small-molecule prenyl inhibitor NHTD (Table 3) blocks the prenyl-binding pocket as well as localization to the plasma membrane, resulting in the disruption of oncogenic *KRAS*-PDEδ binding [102]. NHTD also can induce apoptosis in different types of lung cancer cells. In the in vivo study, NHTD inhibited tumor progression in xenograft and mouse models by decreasing CRAF, ERK, and AKT phosphorylation. Computational molecular docking studies identified a specific binding site for a particular small molecule.

#### 4.3.2. PD98059

It was reported that colon cancer is associated with two genetic events: the Wnt signaling pathway and *KRAS* mutation. Additionally, vascular endothelial growth factor (VEGF) is operated by both genetic factors [128]. *KRAS* exerts an oncogenic effect by activating multiple Raf/MEK/ERK and PI3K pathways. An experiment was conducted with LY294002 and PD98059 (Table 3) on the ERK and PI3K pathways to determine if the Wnt signaling pathway follows similar pathways for the association of colon cancer. The results indicated that the PI3K effector pathway was critical for the *KRAS*^val12^-mediated stimulation of Wnt signaling.

Additionally, further investigation was conducted to determine if both *KRAS* and Wnt signaling regulate the phosphorylation of TCF4. The results showed that β catenin formed a complex in both the phosphorylated and unphosphorylated form, but *KRAS* did not control the phosphorylation of TCF4. β catenin is the key regulator for Wnt signaling; *KRAS* regulates the stability of β catenin. Moreover, *KRAS* controls GSK-3 β Activity independently of Serine-9 Phosphorylation. Finally, these two genetic factors follow a unique interaction between two oncogenic pathways by which *KRAS* increases signaling via the Wnt pathway in colon cancer [128].

#### 4.3.3. Wortmannin

Wortmannin (Table 3) is a PI3K inhibitor that covalently binds to the p110 subunit of PI3K; it irreversibly inhibits PI3K and decreases AKT phosphorylation [131,132]. Bialkowska and colleagues [129] identified that wortmannin decreased pAKT levels in DLD-1 cells via a reduction in KLF5 levels. Recent studies found that PI3K/AKT activation is often associated with colorectal cancer and enhanced colorectal cancer development [133]. Wortmannin inhibits colorectal cancer by suppressing the upregulation of PI3K in a dose-dependent manner.

### 4.4. Mixed Heterocyclic Small-Molecule RAS Inhibitors

According to the structure, mixed heterocyclic small-molecule RAS inhibitors are classified in Table 4. The bio-molecular targets and mechanistic studies are also described.

#### 4.4.1. Talniflumate + Gefitinib

Mucin is one of the major culprits that hinder drug delivery, and several clinical studies identified that mucin is overexpressed in *KRAS*-driven pancreatic cancer in mouse and human models [140,141,142,143]. Enzyme 2 β-1,6 N-acetylglucosaminyltransferase (GCNT3) is recognized as a novel core mucin-synthesized enzyme and targeting this enzyme could decrease the overexpression of mucin [134]. *KRAS* mutation with p48^Cre/+^-LSL-*KRAS*^G12D/+^ GEM upregulates the mucin concentration in pancreatic intraepithelial neoplasia (PanIN) and PDAC. Further study found that GCNT3 enzyme is abnormally expressed from GEM in pancreatic cancer as compared to the pancreases under normal conditions, resulting in high mucin formation. Therefore, GCNT3 is used as a novel target to inhibit the overexpression of mucin in pancreatic cancer. Talniflumate (Table 4) is a mucin inhibitor with a good binding capacity with GCNT3 following in-silico validation. The docking score of talniflumate is very impressive compared to the known ligand GALB1, 3GALNAC. Further study confirmed that talniflumate, after binding with GCNT3, inhibits the protein expression of GNCT3 and significantly decreases the overexpression of mucin. Further investigation of the EGFR inhibitor (gefitinib) (Table 4) confirmed the occurrence of remarkable tumor regression in PDAC and PanIN, with a significant decrease in mucin expression.

#### 4.4.2. CPD0857 and KY1022

Wnt/β-catenin and RAS-MAPK signaling are the central pathways for *KRAS*, and APC mutation causes drug-resistant colorectal cancer (CRC) [144]. Recently, Jung Kyu Choi and colleagues [135] identified a novel compound with inhibitory capacity against both the Wnt/β-catenin and RAS-MAPK signaling pathways in drug-resistant CRC. Mechanistically, CPD0857 (Table 4) does not downregulate β-catenin and RAS protein signaling, indicating that it might follow a different mechanism. After observing the apoptotic activity of CPD0857, the results suggested that the inhibition of cell growth and HCT116 happened due to PI3K/AKT suppression, and in vivo tumor suppression occurred following the inhibition of Wnt/β-catenin and RAS/ERK signaling [135]. A previous study found that another small-molecule inhibitor, KY1022 (Table 4), destabilizes both Wnt/β-catenin and RAS/MAPK signaling. This study explicitly targeted KRAS^G12D^ in a mouse model, and the results showed a significant decrease in tumor growth [145].

#### 4.4.3. KYA1797K

Another small-molecule KYA1797K (Table 4), was identified as a potent inhibitor of cetuximab-resistant *KRAS*-driven colorectal cancer [137]. This small-molecule inhibitor functions by disrupting β-catenin and RAS via GSK3β activation. KYA1797K overcame the cetuximab-resistant *KRAS* mutation when applied to the cetuximab-resistant CRC cell, but it showed dose-dependency in inhibiting the mutation. In vivo analysis was also performed to observe the cetuximab-resistant *KRAS*-harboring CRC tumor in the Xenograft mouse model [137]. There was no tumor suppression when cetuximab was solely applied to the mouse model. On the other hand, KYA1797K alone or combination with cetuximab showed significant regression of tumor growth. A further experiment conducted on an ApcMin/+/*KRAS*^G12DLA2^ mouse model showed tumor growth inhibition, but also demonstrated cetuximab resistance [137].

#### 4.4.4. 0375-0604

Xie et al. [146] synthesized a new small-molecule inhibitor, 0375-0604 (Table 4), which exerts a binding effect with the switch region (switch-I, II) of the *KRAS*-harbored cell line. The significant characteristic of 0375-0604 is that it comprises two hydrogen bonds that bind with the backbone of Met67 and the side chain of Glu37 in the switch I and switch II pocket, respectively. Moreover, after binding to the pocket, 0375-0604 inhibits *KRAS* mutation by downregulating RAF/MEK/ERK and RAF/PI3K/AKT signaling pathway. It induces apoptosis, leading to the arrest of the G2/M cell cycle in the *KRAS*-harbored NSCLC cell line.

#### 4.4.5. 7773

Oncofetal RNA-binding protein (Igf2bp1) is responsible for different types of cancer [138] and can synergize mutation to KRAS. Researchers synthesized the small-molecule inhibitor 7773 (Table 4), which can bind with the hydrophobic surface of Igf2bp1 in the KH3 and KH4 domain and inhibit the binding of KRAS 6 RNA in vitro. Targeting Igf2bp1, this small-molecule inhibitor can reduce KRAS mRNA, resulting in a decrease in KRAS protein and downregulation of signaling, thereby decreasing cell growth. Thus, it might be a potential therapeutic target in cancer treatment.

#### 4.4.6. NSC-658497

RAS signaling is activated by SOS1 (guanine nucleotide exchange factors) [139], which is responsible for receptor tyrosine kinase signaling to RAS. A small-molecule inhibitor, NSC-658497 (Table 4), was identified as interacting with SOS1, competitively inhibiting SOS1–RAS interaction, and suppressing SOS1 GEF activity following dose-dependency. Mechanistically, the structure of NSC-658497 comprises aromatic benzopyran and the polar nitrophenyl moiety. A benzopyran derivative interacts with the hydrophobic pocket of SOS1, which acts as bait within the active site of SOS1. In contrast, the polar nitrophenyl moiety may interact with the outside of the hydrophobic pocket of SOS1. Following this mechanism, NSC-658497 showed an inhibitory effect against RAS oncogenic protein. NSC-658497 was also found to inhibit cell proliferation in a murine model by downregulating the pERK1/2 and pAKT signaling pathways.

#### 4.4.7. JNJ-74699157

JNJ-74699157 (Table 4) was identified as a *KRAS*^G12C^ inhibitor and was found to covalently bind to *KRAS*^G12C^ at the critical mutant residue cysteine 12, locking the GTPase in an inactive state and inhibiting *KRAS* signaling [147]. This small-molecule inhibitor is now in a phase-I clinical trial. The clinical trial of JNJ-74699157 was reported to be completed within one year, with only 10 participants and no results published [148].

### 4.5. Carbocyclic Small-Molecule RAS Inhibitors

Carbocycles are also very important as heterocycles in drug discovery research. After meticulous investigation, small-molecule RAS inhibitors with the carbocyclic structure are shown in Table 5.

#### 4.5.1. PKF115-584 (Calphostin C)

Combination therapy of PKF115-584 (Table 5) (β-catenin inhibitor) and transfarnesylthiosalicylic acid (FTS, salirasib) (RAS inhibitor) can inhibit both Wnt-associated and *KRAS*-associated colorectal cancer [154,155]. Although both inhibitors can solely suppress colorectal cancer, combination therapy synergizes their inhibitory effect. The investigation found that when this combination drug therapy was applied to two colorectal cancer cells carrying both β-catenin and the *KRAS* mutation, the Ls174T cells showed inhibition of MAPK signaling while the DLD-1 cells inhibited FOS expression, which downregulates RalA [149]. This combination therapy also showed significant cell growth-inducing apoptosis, which occurred because of the downregulation of survivin activity. In addition, a combination of these two inhibitors suppressed tumor growth, and their synergistic effect was specific to the *KRAS* and Wnt mutant as they did not show any effect on wild-type *KRAS*. It was also identified that treatment with these combined inhibitors requires a small dose and exerts minimal side effects [149].

#### 4.5.2. Kobe0065 + Kobe2602

Identification of a suitable or specific binding pocket on the surface of the RAS proteins where the therapeutically active small-molecule drug can bind is a promising strategy for treating RAS-related cancer. Thus, Shima et al. [150] identified a therapeutic family of small-molecule inhibitors (Kobe0065 and its analog Kobe2602) (Table 5) that can inhibit multiple types of RAS mutation. A previous study found that small-molecule inhibitors can inhibit SOS expression by binding with *KRAS*-GDP [156]. After in silico validation, it was identified that Kobe0065 and Kobe2602 could inhibit RAS activity both in vitro and in vivo, binding with HRAS·GTP-c-Raf-1. Both inhibitors efficiently downregulated the phosphorylation of MEK and ERK. These downregulations inhibited the HRAS^G12V^ mutation in NIH 3T3 cells. In addition, they showed antitumor activity on *KRAS*^G12V^ mutated SW480 colon cancer cells when orally administered.

#### 4.5.3. 3,3’-(Ethylazanediyl)bis(N-phenylpropanamide)

Methylation of the carboxy-terminal amino acid is one of the post-transcriptional modifications of oncogenic RAS protein that is activated by isoprenylcysteine carboxylmethyltransferase (ICMT) [157]. The inhibition of isoprenylcysteine carboxylmethyltransferase (ICMT) can be an effective pathway for RAS-associated cancer treatment. Marín-Ramos et al. [151] identified a potent small-molecule inhibitor (Table 5) that can impair four RAS isoforms (HRAS, *KRAS*4A, and -4B, and NRAS). The activity of compound 3 was tested on *KRAS* (G12C, G12D, G12V, and G13D) isoforms and two NRAS-mutated cell lines (significantly found in myeloid leukemia and myeloma [158]); compound 3 demonstrated significant cytotoxic activity for all the isoforms, with the most potent IC_50_ value being observed in the NRAS (Q61K) isoform [151].

#### 4.5.4. Salirasib + FTS, Salirasib

HRAS upregulation is frequently identified in bladder cancer (BC) [159]. Salirasib (Table 5) has been identified as a potent RAS inhibitor in different types of cancer but requires a high concentration for inhibition. Sugita et al. [152] investigated the therapeutic activity of salirasib in HRAS-mutated BC. The authors investigated the binding of salirasib with siRNA targets HRAS, and it was found to result in the inhibition of cell proliferation invasion. Proteomic analysis showed that salirasib inhibits the glycolysis and oxidative phosphorylation pathways. Another series comprising the RAS inhibitor S-trans,trans-farnesylthiosalicylic acid (FTS salirasib) (Table 5), which interacted with the RAS membrane, showed a transformation. Goldberg and co-workers developed [153] a new derivative of FTS salirasib where the carboxyl group was modified by esterification and amidation. They identified all the modified FTS salirasib amides, and it was found that two esters significantly inhibited the growth of Panc-1 and of U87 cells, where at least one RAS isoform showed a chronic effect. They also identified that the modified FTS salirasib did not produce any toxicity in Panc-1 and U87 cells.

### 4.6. Miscellaneous Small-Molecule RAS Inhibitors

There are some small-molecule RAS inhibitors that are in clinical trials. Details about these RAS inhibitors have not been published yet. We also identified one thioheterocyclic small molecule RAS inhibitor. In Table 6, we classified these RAS inhibitors as miscellaneous small-molecule RAS inhibitors.

#### 4.6.1. ML264

Krüppel-like factor 5 (KLF5) is a transcription factor that is overexpressed in proliferating intestinal crypt epithelial cells. KLF5 acts as a mediator of the RAS/MAPK and WNT signaling pathways under homeostatic conditions, and it increases their oncogenic function in intestinal adenomas. Sabando et al. [160] identified a novel KLF5 inhibitor with a therapeutic effect in colorectal cancer. Treatment with ML264 (Table 6) in DLD-1 and HCT116 colorectal cell lines showed a regression in the protein levels of KLF5 that paralleled a reduction in the levels of the transcription factor early growth response 1 (EGR1), which is a direct effector of KLF5 expression. The application of ML264 in a nude mice model inhibited proliferation in colorectal cancer.

#### 4.6.2. GDC-6036

GDC-6036 is a c inhibitor. The combination of GDC-6036 with atezolizumab causes intracellular interactions (ICI), its combination with cetuximab or erlotinib blocks EGFR, and the combination of GDC-6036 with bevacizumab inhibits VEGF. GDC-6036 is also in phase I clinical trial. However, the structure of this compound is not disclosed [148].

#### 4.6.3. LY3499446

LY3499446 is another *KRAS*^G12C^ inhibitor that is combined with abemaciclib, and it inhibits the regulation of CKD4. This small-molecule inhibitor showed promising results in phase I clinical trials. Unfortunately, in 2020, LY3499446 was terminated due to unexpected toxicity [148,161].

#### 4.6.4. D-1553

D-1553 also inhibits *KRAS*^G12C^ mutants. This inhibitor is in a phase I clinical trial. Details about these inhibitors have not been published, and data from ongoing clinical trials have not been reported [148,161].

## 5. Small-Molecule Natural Products as RAS Inhibitors

Various anticancer compounds have been identified from natural sources. Researchers identified that from 1981 to 2019, the US federal drug administration (FDA) approved 185 small-molecule anticancer drugs to treat various types of cancer. Among these 185 small-molecule anticancer drugs, 43% are from natural product derivatives, and 18% are unaltered natural product anticancer drugs [162,163,164]. These data highlight that natural products comprise a vast research area in cancer treatment. Herein, we describe natural product RAS inhibitors and their molecular mechanisms of action.

### 5.1. Natural Product RAS Inhibitors with Heterocyclic Skeleton

#### 5.1.1. Quercetin

Quercetin (Figure 3) is a dietary flavonoid found in tea, onions, grapes, wines, and apples [165]. Investigations found that quercetin can inhibit human colorectal cancer by inhibiting the expression of the p21 RAS mutant. An immunocytochemical study found that the administration of 10 μM quercetin reduces p21 *KRAS* in colon cancer cells and in initial colorectal cancer. This quercetin treatment is time and concentration-dependent. At twenty hours, 10 μM quercetin treatment induces 50% p21 *KRAS* inhibition. Interestingly, quercetin produces a similar therapeutic effect for *KRAS*, HRAS, and NRAS oncoproteins, and their effect does not depend on the cell cycle position of colon cancer [165]. Zhang and co-workers [166] investigated the possibility that quercetin also inhibits the cell growth and metastasis of osteosarcoma. Although quercetin administration solely decreases osteosarcoma, its combination with cisplatin, a well-known chemotherapeutic agent, synergizes with the quercetin activity in osteosarcoma. Additionally, treatment with 5 μM quercetin in 143B cells did not show cell viability, whereas significant cell viability was observed with 10 μM quercetin. Furthermore, when 5 μM cisplatin was administrated to 143B cells, the IC_50_ value was 6.12 M. In the meantime, cotreatment with 5 μM of quercetin and cisplatin showed an IC50 of 4.21, which indicates that quercetin enhanced the cisplatin sensitivity of 143B. Cisplatin showed resistance towards osteosarcoma, and MiR-217 decreased the resistance of osteosarcoma [167]. Treatment with quercetin and/or cisplatin upregulates MiR-217, but treatment with *KRAS* downregulates mRNA and protein levels.

#### 5.1.2. Artemidolide C

Artemidolide is a dimeric sesquiterpene lactone and FPTase inhibitor. It is an (A-D) series that is isolated from *Artemisia* spp. [168]. In vitro, artemidolide C (Figure 3) treatment showed dose-dependent inhibition, and approximate growth inhibition followed the administration of 1.3–8.1 μM. The growth inhibition in three human cancer cell lines (colon, breast, and CNS) showed sensitivity when treated with artemidolide C, but the renal tumor cell line A498 showed resistance [168]. An earlier study identified that, due to the presence of the α-methylene-γ- lactone group, sesquiterpene lactones exhibit antitumor activity [169]. However, the α-methylene-γ- lactone unit in artemidolide did not affect the inhibition of FPTase activity, but the problem was solved when hydrogenated artemidolides exerted low antiproliferative activity. Furthermore, in vivo antiproliferative activity in nude mice showed that artemidolide C inhibits tumor growth [168]. Another study showed that treatment with artemidolide C in a HRAS-mutated cell line resulted in degradation of the HRAS protein [170].

#### 5.1.3. Statins

Mevalonate intermediates, including farnesyl pyrophosphate (FPP) and geranylgeranyl pyrophosphate (GGPP), which are responsible for activating RAS proteins, were potentially inhibited by statins in pancreatic cancer [171]. Simvastatin (Figure 3) treatment in MiaPaCa-2 human pancreatic cancer cells showed that 200 genes were affected by simvastatin treatment. This was due to the interaction between FPP and simvastatin. However, it was observed that the normalization of the expression of *KRAS*-related genes and the GFP-*KRAS* protein trafficking was partially prevented by the addition of any of the mevalonate pathways intermediates. Finally, the addition of FPP or GGPP normalized simvastatin-treated altered genes. Therefore, *KRAS* protein trafficking can be successfully inhibited by statin treatment in pancreatic cancer [171].

#### 5.1.4. Manumycin A

Exosomes are necessarily involved in the trafficking of oncogenic factors and the neoplastic alteration of stem cells, resulting in cancer progression [172,173]. Manumycin A (Figure 3) (MA) is a natural macrolide antibiotic isolated from *Streptomyces parvulus* and has been identified as an exosome that can be targeted therapeutically. To determine the nontoxic dose of manumycin A and evaluate the effect of exosome biogenesis, MA was applied to RWPE-1 and PC-3 cells, and no effect was observed, while approximately 8% and 10% cell death was observed on C4-2B and 22Rv1 cells, respectively [174]. However, using the EV analysis based on TRPS, the Nanosight300 and NanoFACS analyses reported that MA selectively inhibits the biogenesis and secretion of exosomes in some castration-resistant prostate cancer (CRPC) cells. In addition, MA inhibited exosome biogenesis and secretion by downregulating the RAS/Raf/ERK1/2 pathway in CRPC cells. Finally, MA inhibits exosome biogenesis and secretion, suppressing RAS signaling pathways (RAS/Raf/ERK).

#### 5.1.5. Gliotoxin

Liver injury happens due to an overdose of analgesic and antipyretic acetaminophen (APAP). Article [175] described that hepatotoxicity due to an excessive dose of APAP activates RAS mutation in mice models. Gliotoxin (Figure 3) is a sulfur-containing mycotoxin produced by various pathogenic fungi, including *Aspergillus fumigatus* [176]. As a potent farnesyl transferase inhibitor (FTI), Gliotoxin decreased RAS overexpression upon APAP overdosing. It was also identified that RAS activation concomitantly increased with hepatotoxicity [175].

Furthermore, APAP dosing reduces the hepatic glutathione amount, but manumycin A (inhibiting the RAS GTP and ALT interaction) treatment does not affect this. Additionally, the results suggested that RAS activation is regulated by JNK phosphorylation because APAP dosing induces JNK phosphorylation, and the application of manumycin A significantly decreases JNK phosphorylation. After APAP dosing, gliotoxin treatment reduces serum amounts of ALT and IFN-γ and inhibits RAS activation [175].

#### 5.1.6. Preussomerin G

In 1991, Holly A. and colleagues isolated a group of preussomerins (A–F) (preussomerin G, Figure 3) from the coprophilous fungus *Preussia isómera* [177]. Farnesyl protein transferase is responsible for RAS farnesylation (p21), which is associated with the RAS plasma membrane and helps to produce RAS signaling. Singh et al. [178] isolated preussomerins and deoxipreussomerins from the extract of a dung-inhabiting coelomycetous fungus from Chaco Province, Argentina, which has inhibitory function towards RAS farnesylation.

#### 5.1.7. Pepticinnamin E

Pepticinnamin E (Figure 3) is another FPTase inhibitor. A series of Pepticinnamins were isolated from the culture broth of *Streptomyces* sp. OH-4652(18). Among them, pepticinnamin E is the primary product that contains a rare N-terminal cinnamoyl moiety and several nonproteinogenic amino acids. All isolated pepticinnamin showed a potent inhibitory property towards FTase with IC_50_ values 0.1–1.0 μM, but pepticinnamin C produced the strongest inhibition. Furthermore, pepticinnamin E exerts an effect of competitively binding with the p21 RAS protein and non-competitively binding to the substrate farnesyl pyrophosphate [179]. Another study identified a strong reduction in FPTase E, which is expected to treat cancer and malaria [180]. Thutewohl et al. [181] found pepticinnamin E induces apoptosis in RAS-mutated cell lines, which is connected to the inhibition of FTase activity. They identified 51 analogs of pepticinnamin. Overall, twenty compounds showed an inhibitory effect with an IC_50_ value of 1 μM.

#### 5.1.8. Bryostatin-1

Protein kinase C (PKC) phosphorylates S181 into the polybasic region that promotes the rapid dissociation of *KRAS* in the plasma membrane and has an association with the outer membrane of mitochondria where phospho-*KRAS* interact with Bcl-X_L_. The PKC agonist accelerates the apoptosis of cells altered with the *KRAS* oncogene. *KRAS*, with a phosphomimetic residue at position 181, induces apoptosis via a pathway that requires Bcl-XL [182]. Bryostatin-1 (Figure 3) is a cyclic macrolide that is isolated from the marine bryozoan *Bugula neritina* [183]. Moreover, Bryostatin-1 is a PKC agonist [184] that inhibits *KRAS*-dependent cell transformation and growth in an S181-dependent manner. An in vivo study showed that bryostatin-1 was efficient against tumors in nude mice containing oncogenic KAS^G12V^, but in *KRAS*^G12V181A^-driven tumors, it had reduced activity [182].

#### 5.1.9. Piperlongumine

Piperlongumine (PL) (Figure 3), a natural alkaloid present in *Piper longum* Linn, has been reported to exhibit notable anticancer effects in various in vitro studies. Kumar and colleagues [185] first identified a chemo-preventive effect of PL in colon cancer-infected animal models. PL showed significant antineoplastic activity towards colon cancer cell growth by targeting RAS proteins and the PI3K/Akt signaling pathway. It was also identified that PL blocked the cell cycle progression at the G2/M phase and increased the mitochondrial apoptotic pathway by downregulating Bcl-2 levels. Moreover, PL showed liver and kidney toxicity.

#### 5.1.10. Confluentin

Yaqoob et al. [186] isolated three compounds, confluentin (Figure 3), grifolin, and neogrifolin, from a methanolic extract of the terricolous polypore of *Albatrellus flettii* (British Columbian mushroom), which showed cell viability. However, the cell viability mechanism of this novel molecule is not very well established. In a cytotoxic study, the confluent showed IC_50_ values of 25.9 ± 2.9 μM (*n* = 3), 33.5 ± 4.0 μM (*n* = 3), and 25.8 ± 4.1 μM (*n* = 3) against HeLa, SW480, and HT29 cells, respectively [186]. Moreover, they identified confluentin-induced apoptosis for the first time, with a cell-cycle arrest at the G2/M phase in SW480 human colon cancer cells. Additionally, *KRAS*^G12D^, containing the SW480 human colon cancer cell line, was used to investigate the *KRAS* inhibition of these three molecules. Confluentin showed a more significant effect than grifolin and neogrifolin, but in the HT29 cell line, which contains wild-type *KRAS*, a similar inhibitory effect against the *KRAS* mutation was observed for all three compounds. Finally, using an in vitro fluorescence polarization method, it was found that confluentin inhibits the physical interaction between IMP1 and *KRAS* RNA [186].

#### 5.1.11. Swinhopeptolides

Two new cyclic depsipeptides named swinhopeptolides A and B were isolated from the marine sponge *Theonella swinhoei* cf. *verrucosa*, collected from Papua New Guinea [187]. Both compounds have 11 diverse amino acid residues and 13-carbon polyketide moieties attached at the N-terminus. To identify the effect on RAS-Raf protein–protein interactions of Swinhopeptolides A (Figure 3) and B, a cell-based Bioluminescence Resonance Energy Transfer (BRET) was performed. The results showed that swinhopeptolides A and B effectively inhibited RAS-Raf protein–protein interaction with IC_50_ values of 5.8 and 8.5 μM, respectively [187]. Moreover, Swinhopeptolides A and B exhibited potential RAS/Raf signaling pathway suppression.

#### 5.1.12. Avicin G

Avicin G (Figure 3) is a plant-derived triterpenoid saponin from *Acacia victoriae*; it mislocalizes *KRAS*^G12V^ from the Plasma membrane (PM) and disrupts the PM spatial organization of oncogenic *KRAS* and HRAS by depleting their phosphatidylserine (PtdSer) and cholesterol contents, respectively, at the inner PM leaflet [188]. Further investigation suggested that avicin G inhibits *KRAS*^G12V^ and HRAS^G12V^ mutation by downregulating the pERK signaling pathway and pAkt, but more inhibition was observed for *KRAS*^G12V^ cells. Next, a cell proliferation assay was performed on human pancreatic ductal adenocarcinoma (PDAC) cells and non-small-cell lung carcinoma (NSCLC) cells. The results showed significant growth inhibition for both types of the cancer cell line. Previous research identified that lysosome activity is vital for *KRAS*-driven cancer growth, but avicin G [189,190] blocks the lysosome activity by elevating lysosomal PH and inhibiting the phosphorylation of ERK signaling. It was also identified that Avicin G is a potent SMase inhibitor. It exerts a potential inhibitory effect on acid and neutral SMase [188].

### 5.2. Natural Product RAS Inhibitor with Carbocyclic Skeleton

#### 5.2.1. Prostratin

Prostratin (Figure 4) is a phorbol ester, which was first isolated from the bark of the mamala tree of Samoa*, Homalanthus nutans* (Euphorbiaceae) [191]. Prostratin inhibits *KRAS* and HRAS tumor growth by suppressing non-canonical Wnt/Ca^2+^ signaling. Comparing *KRAS* and HRAS, *KRAS* produces more tumorigenesis, although they have a comparable level of canonical RAS signaling. The reason for this is the ability to induce tumor initiation that is directly related to the ability of *KRAS* to suppress Fzd8-mediated non-canonical Wnt/Ca^2+^ signaling. In the *KRAS*^G12V^ mutation, Fzd8 was downregulated, and the CaMKii level was decreased more significantly than that of HRAS^G12V^, resulting in the phosphorylation of threonine 286. Therefore, prostratin (PKC activator) interrupts *KRAS* calmodulin-binding that increases the level of Fzd8, thereby inhibiting *KRAS* mutation in pancreatic cancer cells [191].

#### 5.2.2. D-Limonene

Farnesyl protein transferase (FPTase) is an intermediate of the mevalonate pathway, which contains the isopropyl group. The interaction of FPTase with a potent inhibitor can repress malevolent pathways, resulting in the inhibition of transforming characteristics of the RAS mutant. D-Limonene (Figure 4) is a common monoterpene that can inhibit FPTase, and has been identified in essential oils of orange, lime, grapefruit, lemon, mandarin, and many other plants. To determine the FTPase inhibition activity, D-Limonene and some other compounds were applied to partially purified FPTase from LLC-PK 1 cells [192]. The results suggested that among all the tested compounds, D-Limonene, TD-I, TD-I1, and gallotannin produce an inhibitory effect towards FPTase but in a dose-dependent manner. To identify the cell proliferation activity, D-Limonene, TD-I, TD-I1, and gallotannin were tested on W1-38, CACO, A549, and PaCa cells; significant cell proliferation was identified for all the cell lines and the effects on PaCa cells were due to the inhibition of P21^RAS^ membrane association. Additionally, the administration of D-Limonene on A549 cells showed moderate HRAS inhibition. Moreover, western blot analysis showed that D-Limonene also has an increased cytotoxic effect [192].

Another study found that D-Limonene significantly affected skin tumor growth in mouse models by inhibiting 12-*O*-tetradecanoylphorbol-13-acetate (TPA) [193]. The development of mouse skin tumor was initiated with 7,12-dimethylbenz[a]anthracene (DMBA) and went to the progression state because of 12-*O*-tetradecanoylphorbol-13-acetate (TPA). In comparison to TPA treatment [194] (previously studied), topical treatment with D-Limonene (12.30 h) in a mouse model showed only 27% and 9% increases in the weight of the skin paunches, whereas TPA (12 h) showed an increase of 48% [193]. Similarly, D-Limonene treatment significantly reduced COX-2-positive cells in the epidermis. Additionally, the application of D-Limonene suppresses the RAS, Raf signaling, and phosphorylation of extracellular protein kinase in the case of DMBA-/TPA-induced tumors [193].

#### 5.2.3. Methyl Linderone

Methyl linderone (ML) (Figure 4) is a cyclo-pentenedione isolated from the fruits of *Lindera erythrocarpa* (family Lauraceae). Yoon et al. [195] identified that treatment with ML inhibits breast cancer. After performing a cell viability assay, they found that 10 μM is the nontoxic dose of ML. To study the effect of ML in invasion and migration, 12-*O*-tetradecanoyl phorbol-13-acetate (TPA)-stimulated MCF-7 cells were used. The finding was that TPA-stimulated invasion and migration were inhibited by MA, but in a dose-dependent manner. Furthermore, it was identified that ML inhibited two mRNA—MMP-9 and IL-8—and the protein expression in TPA-treated MCF-7 cells. Moreover, the mechanism of inhibition of MA on MCF-7 cells was followed by the targeting of the TPA-stimulated phosphorylation of ERK and the downstream factors, AP-1 and STAT3.

#### 5.2.4. Antroquinonol

Antroquinonol (Figure 4) extracted from the mycelium of *Antrodia camphorate*, was identified as the smallest anticancer molecule [196]. Although the mechanism of action of Antroquinonol remains unclear, some evidence suggested that antroquinonol inhibits RAS- and RAS-mediated protein function by blocking the expression of isopropyl protein transferase. Moreover, it was identified that antroquinonol selectively binds with FPTase and geranyl transferase 1, the most RAS-activating enzyme. Antroquinonol effectively inhibits the RAS activity in vitro. It was also determined that antroquinonol can induce apoptosis by blocking the PI3K/mTOR pathway [197]. Moreover, antroquinonol treatment in lung cancer cells induces autophagic property of LC3B-1 to LC3B-II.

#### 5.2.5. 5-CQA (5-*O*-Caffeoylquinic Acid)

Chlorogenic acid (5-*O*-caffeoylquinic acid, 5-CQA) was identified in coffee beans used to make green coffee and, after roasting, black coffee, a widespread drink worldwide. 5CQA (Figure 4) was the first natural compound targeting RAS mutation [198]. A current study found that intake of healthy food, abundant in polyphenols, decreases neurovegetative diseases, with 5-CQA being one of them [199]. Moreover, 5CQA affects inflammation, cancer, and neurovegetative diseases. Although the molecular mechanism of biological effectiveness of 5CQA is not investigated, an investigation found that it inhibits RAS-dependent breast cancer. After evaluating the standard NMR result, it was identified that 5-CQA reduces cancer growth by modulating p42/p44 MAPKs activation/phosphorylation. A molecular docking study found four binding sites (S-I-P and S-II-P for both complexes, and S-I-II-P for the HRASGDP complex only) in oncogenic HRAS protein where 5-CQA can bind and determined the FullFitness scores. It was also identified that 5-CQA noncompetitively interacted with the binding site because the computationally found 5-CQA binding site was distinct from the active site.

#### 5.2.6. Lupeol

Lupeol (Figure 4) is a triterpenoid, a FPTase inhibitor, and specifically inhibits *KRAS* mutants, but not wt *KRAS* [200]. After analyzing its crystal structure, Ganaie et al. identified 20 terpenoids for their *KRAS*-binding affinity. In a comprehensive study on lup-20 (29)- en-3*b*-ol (lupeol) as a *KRAS* inhibitor, differential scanning fluorimetry, immunoprecipitation assays, isothermal titration calorimetry were performed, and lupeol was identified as a potent *KRAS* inhibitor. Lupeol has a potential inhibitory effect on mutant *KRAS*^G12V^. In vivo lupeol administration in a mouse model showed inhibition of the development of pancreatic intraepithelial neoplasia.

#### 5.2.7. Grifolin and Neogrifolin

Grifolin (7a) and neogrifolin (7b) (Figure 4) were isolated from the ethanolic extract of a bioactive mushroom that is indigenous to British Columbia [186]. One more compound, confluentin, was isolated. We discussed the biological activity of confluentin in an earlier section. The aforementioned investigation showed that grifolin and neogrifolin could inhibit oncogenic *KRAS* in human colon cancer. They also exerted their inhibitory effect in the presence of wild *KRAS* [186].

### 5.3. Natural Product RAS Inhibitor with Acyclic Skeleton

#### Chaethomellic Acids A

Chaethomellic acids are a class of alkyl dicarboxylic acids isolated from *Chaetomella acutiseta* [201]. An investigation identified two novel compounds, chaethomellic acid A (Figure 5) and B, identified three structural types of FPTase inhibitors—phosphonic acids, chaethomellic acids, and zaragozic acids—that produced weak inhibition to geranyl protein transferase, but each of them showed competitive effects towards farnesyl diphosphate when inhibiting FPTase. Among these three classes, phosphonic acid and chaethomellic acid showed outstanding selectivity towards FPTase and geranyl transferase activity, while zaragozic acids showed less selectivity. Mechanistically, chaethomellic acid A competitively inhibits FPTase and noncompetitively inhibits RAS protein. To identify the FPTase activity, all the compounds were assayed in Ha-RAS containing NIH3T3 fibroblast cells; chaethomellic acids and zaragozic acids had no effect in this assay.

Nogueira et al. [202] identified that long-term treatment with chaethomellic acid A reduces renal mass reduction-induced renal fibrosis in a rat model by selectively inhibiting Ha-Ras farnesylation. After analyzing the previous studies [203,204], it was hypothesized that chaethomellic acid A treated CKD by altering the Raf/MEK/ERK and PI3K-Akt pathways. Chaethomellic acid A was administered in three different rat models (23 µg/Kg), three times a week for 6 months, resulting in significantly decreased renal fibrosis, mainly glomerulosclerosis and arteriolosclerosis.

## 6. Conclusions

Up- and down-regulations of proteins are crucial in terms of causing diseases, including cancer. RAS family proteins, particularly KRAS, play central roles in different stages of cancer development, invasion, propagation, metathesis, and angiogenesis, leading to the formation of solid tumors [205,206,207]. Previously, there was a myth that *KRAS* is undruggable, but continuous efforts from different parts of the world have established that it is druggable. Despite the progress, the current protocols encounter several limitations, including the loss of selectivity, the causing of toxicity, and the production of incongruent patient outcomes. Various genomic and histologic mechanisms promote resistance to covalent *KRAS*^G12C^ inhibitors. For example, the development of heterogeneous responses to some KRAS inhibitors, such as MRTX849 or AMG510, bypass their anti-growth effects by producing more *KRAS* variants, to which these inhibitors do not bind. The developed *KRAS* alterations include G12D/R/V/W, G13D, Q61H, R68S, H95D/Q/R, Y96C, and high-level amplification of the *KRAS*^G12C^ allele. The attained bypass mechanisms of resistance include *MET* amplification, activating mutations in *NRAS*, *BRAF*, *MAP2K1*, and *RET*, oncogenic fusions involving *ALK*, *RET*, *BRAF*, *RAF1*, and *FGFR3*, and loss-of-function mutations in *NF1* and *PTEN*. Accordingly, more in-depth research is required to overcome the resistance towards KRAS inhibitors. In conclusion, in spite of there being unceasing progress, many miles remain in the development of RAS inhibitors with higher potency and selectivity, lower toxicity, and apprehended drug-resistance.

## Figures and Tables

**Figure 1 ijms-23-03706-f001:**
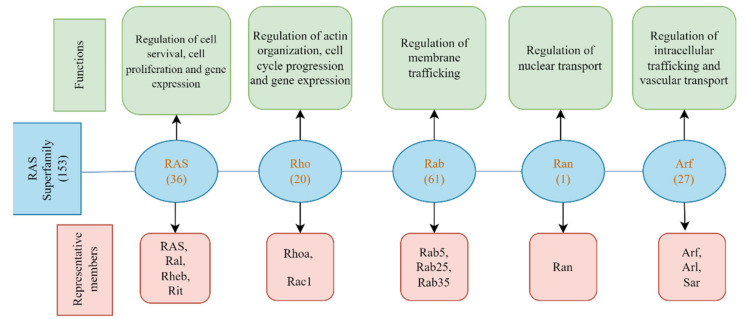
Classification of the RAS superfamily [6,11].

**Figure 2 ijms-23-03706-f002:**
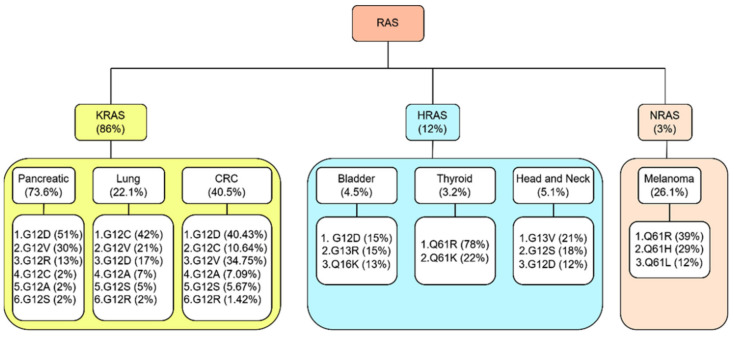
Frequency of *KRAS*, HRAS, and NRAS in different types of cancer [2,50,59,60].

**Figure 3 ijms-23-03706-f003:**
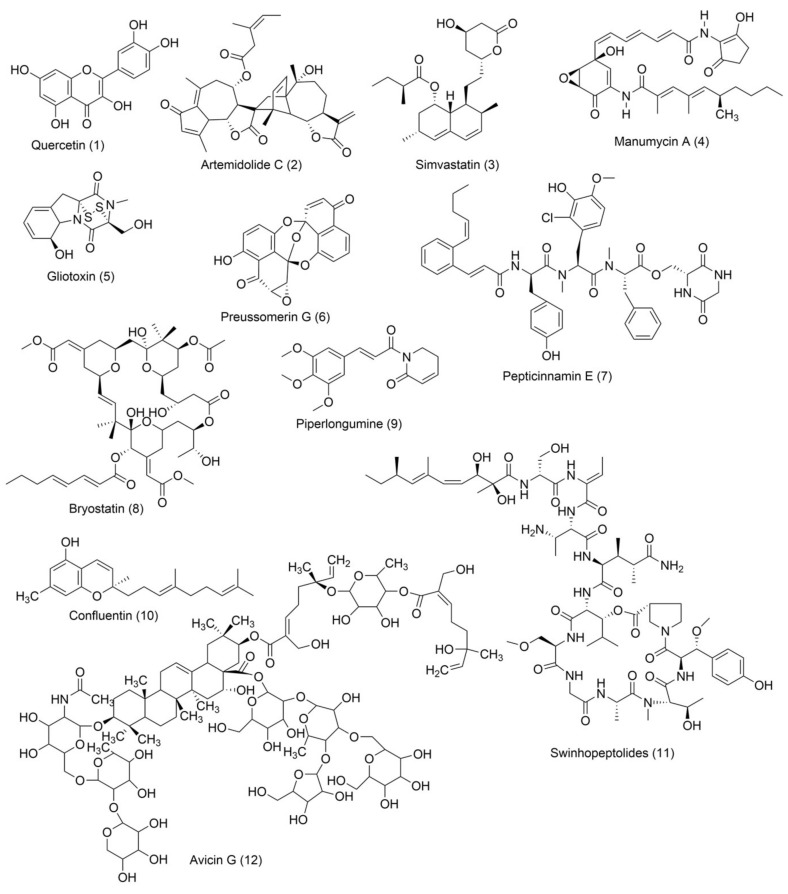
Heterocyclic natural product RAS inhibitors.

**Figure 4 ijms-23-03706-f004:**
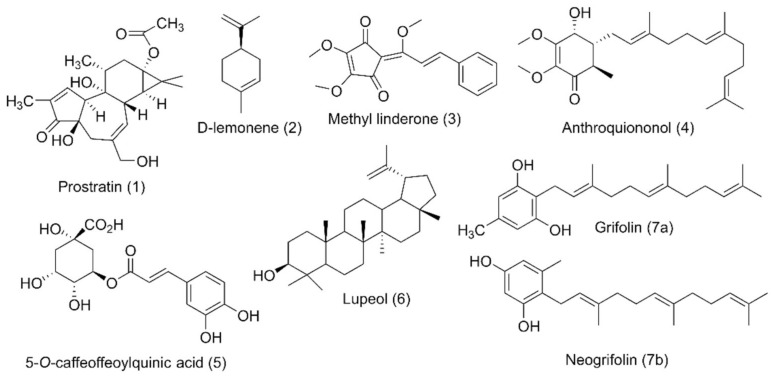
Carbocyclic natural product RAS inhibitors.

**Figure 5 ijms-23-03706-f005:**
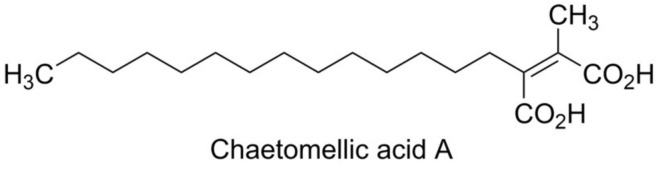
Acyclic natural product RAS inhibitor Chaetomellic acid A.

**Table 1 ijms-23-03706-t001:** Aza heterocyclic small-molecule RAS inhibitors with one nitrogen atom.

Compound Name	Structure	Major Pharmacophore(s)	Cancer Type	Targeted Enzyme M/A	References
Ganetespib	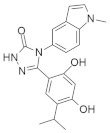	Indole, Triazolone, Resorcinol	Lung cancer/colorectal cancer	Inhibits heat-shock protein 90 (Hsp90)	[74] [75]
Apatinib	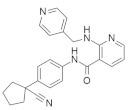	Pyridine	Metastatic lung cancer/breast cancer/gastric cancer	Inhibits vascular endothelial growth factor receptor-2 (VEGFR-2)	[76]
Oncrasin-1	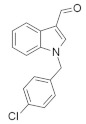	Indole, Chlorophenyl	Lung cancer	Inhibits K-RAS/PKCι pathway	[77]
†	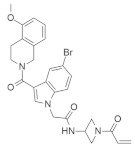	Indole, Azetidine, Isoquinoline	Unknown	Inhibits *KRAS*^G12C^ mutation	[78]
††	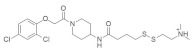	Piperidine, Disulfide linkage	Unknown	Inhibits *KRAS*^G12C^ mutation	[79]
GDC-0449 (Vismodegib) combined with miRNA	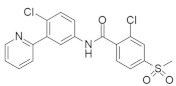	Pyridine, Chlorobenzene, Sulfonyl	Pancreatic cancer	Hedgehog (Hh) inhibitor	[80]

† = *N*-(1-acryloylazet-id-in-3-yl)-2-(5-bromo-3-(5-methoxy-1,2,3,4-tetrahydroi-soquino-line-2-carbonyl)-1*H*-indol-1-yl) acetamide; †† = 2-((4-((1-(2-(2,4-dichlorophenoxy) acetyl) piperidin-4-yl) amino)-4-oxobutyl) disulfaneyl)-*N*,*N*-dimethylethan-1-aminium.

**Table 2 ijms-23-03706-t002:** Aza heterocyclic small-molecule RAS inhibitors with more than one nitrogen atom.

Compound Name	Structure	Major Pharmacophore(s)	Cancer Type	Targeted Enzyme (M/A)	References
ARS-1620	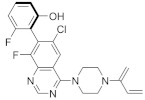	6-chloro-8-fluoroquinazoline, Piperizine	Unknown	*KRAS*^G12C^ inhibitor	[90]
ARS-853	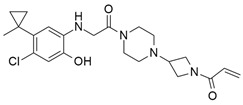	Chlorophenol, Piperazine, Azetidine	Unknown	*KRAS*^G12C^ inhibitor	[91]
AMG 510 (Lumakras or Sotorasib)	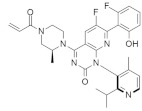	Piperazine, Fluorophenol, Fluoropyrimidinone	Pancreatic cancer/Lung cancer	*KRAS*^G12C^ inhibitor	[92]
MRTX849 (Adagrasib)	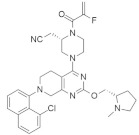	Piperizine, Chloronaphthalene, Tetrahydropyrido[3,4-*d*]pyrimidine	Pancreatic cancer/Lung cancer	*KRAS*^G12C^ inhibitor	[93]
†††	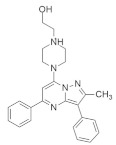	Piperazine, Pyrazolopyrimidine	Pancreatic cancer	Inhibits MAPK/RAF signaling	[94]
BGB324 (Bemcentinib)	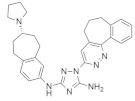	Cycloheptapyridazine, Pyrrolidine, Triazole	Pancreatic cancer	Axl kinase inhibitor	[95]
ABT-737	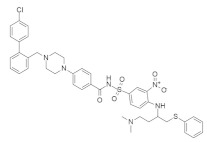	Chlorobiphenyl, Piperazine	Colon cancer	Represses Bcl-2/Bcl-X_L_, resulting in inhibition of the RNAi of PAK4 and PAK1	[96]
AZD6244 (Selumetinib)	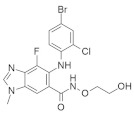	Quinoline, Imidazoquinoline	Colorectal cancer	Downregulation of MEK1/2 pathway inhibits *KRAS* mutation	[97]
NVP-BEZ235 (Dactolisib) in Combination with AZD6244 (Selumetinib)	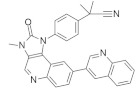	Quinoline, Phenylpropanenitrile, Imidazoquinolinone	Lung cancer	Dual pan PI3K/MEK inhibitor	[98]
R115777 (Zarnestra or Tipifarnib)	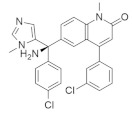	Chlorophenyl, Imidazole, Quinolinone	Myeloma	Inhibits farnesyl transferase signaling	[99]
PPIN-1 PPIN-2	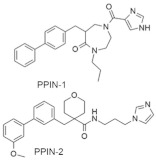	Biphenyl, Imidazole, *N*-propyldiazepanone, Tetrahydropyran	Unknown	PPI inhibitor	[100]
pan-RAS inhibitor 3144 (RAS-IN-3144)	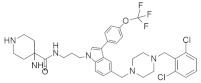	Indole, Piperazine, Trifluoromethoxyphenyl	Unknown	Downregulates PI3K/AKT, RAF/MEK/ERK signaling	[101]
Deltarasin	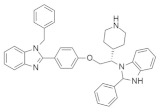	Benzimidazole, Piperidine	Pancreatic cancer/Lung cancer	Downregulates RAS/RAF signaling pathway	[102]
††††	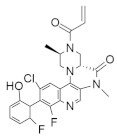	Fluoroquinoline, Fluorophenol, Piperazine, Piperazinone	Unknown	*KRAS*^G12C^ inhibitor	[103]
SML-8-73-1 SML-10-70-1	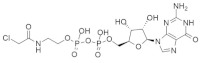 SML-8-73-1 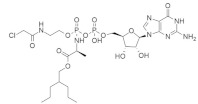 SML-10-70-1	Purine, Tetrahydrofuran	Unknown	Inhibits *KRAS*^G12C^ binding with guanine-binding site	[104]

††† = 1-(2-hydroxyethyl)-4-(2-methyl-3,5-diphenylpyrazolo[1,5-a] pyrimidin-7-yl) piperazin-1-ium. †††† = (2*R*,4*aR*)-3-acryloyl-11-chloro-9-fluoro-10-(6-fluoro-2-hydroxycyclohexa-2,4-dien-1-yl)-2,6-dimethyl-2,3,4,4a-tetrahydro-1*H*-pyrazino [1’,2’:4,5] pyrazino[2,3-*c*] quinolin-5(6*H*)-one.

**Table 3 ijms-23-03706-t003:** Oxo heterocyclic small-molecule RAS inhibitors.

Cancer Type	Compound Name	Structure	Major Pharmacophore(s)	Biomolecular Target (M/A)	Reference
Lung cancer	NHTD	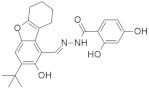	Tetrahydrodibenzofuran, 2,4-Dihydroxybenzohydrazide	Inhibits tumor progression, decreasing CRAF, ERK, and AKT phosphorylation	[102]
Unknown	PD98059	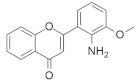	Chromenone	MEK inhibitor	[128]
Colorectal cancer	Wortmannin	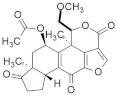	Furoindenoisochromene	Suppresses upregulation of PI3K	[129]

**Table 4 ijms-23-03706-t004:** Mixed heterocyclic small-molecule RAS inhibitors.

Compound Name	Structure	Major Pharmacophore(s)	Cancer Type	Targeted Enzyme (M/A)	Reference
Talniflumate + Gefitinib	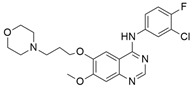 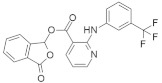	Chlorofluorophenyl, Morpholine, Quinazoline, Pyridine, Benzofuranone	Pancreatic cancer	Inhibition of 2 β-1,6 N-acetylglucosaminyltransferase (GCNT3)	[134]
CPD-0857 and KY1022	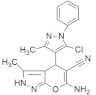 & 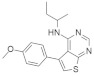	Pyrazole, Pyran, Thienopyrimidine	Colorectal cancer	Inhibition of Wnt/β-catenin, RAS/ERK, and PI3K/AKT	[135]
KYA1797K (ab229170)	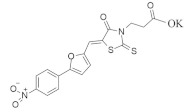	Nitrophenyl, Furan, Thioxothiazolidin-4-one	Colorectal cancer	Unknown	[136]
0375-0604 (*KRAS* inhibitor-9 or DUN09716)	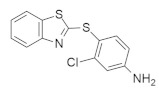	Benzothiazolylthio-3-chloroaniline	Unknown	Inhibition of *KRAS* mutation by downregulating RAF/MEK/ERK and RAF/PI3K/AKT signaling pathways	[137]
7773	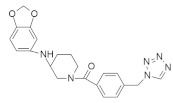	Tetrazole, Piperidine, Benzodioxole	Lung cancer	Binding with hydrophobic surface of Igf2bp1 in KH3 and KH4 domain inhibits *KRAS* mutation.	[138]
NSC-658497	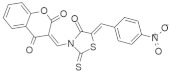	Thioxothiazolidine, Chromane-2,4-dione, Nitrophenyl	Unknown	Binding with hydrophobic pocket downregulates pERK1/2 and pAKT signaling	[139]
JNJ-74699157	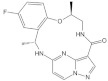	Pyrazolopyrimidine, Benzocyclononaphanone	Unknown	*KRAS*^G12C^ inhibitor	Phase I Clinical trial completed

**Table 5 ijms-23-03706-t005:** Carbocyclic small-molecule RAS inhibitors.

ID	Structure	Major Pharmacophore(s)	Cancer Type	Biomolecular Target (M/A)	Reference
PKF115-584 (Calphostin C)	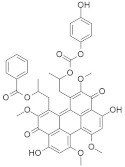	Tetramethoxy-3,10-dioxo-3,10-dihydroperylen-1-yl) propan-2-yl benzoate	Colorectal cancer	Downregulates MAPK and RalA signaling	[149]
Kobe0065 + Kobe2602	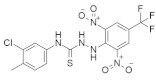 Kobe0065 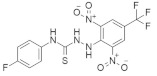 Kobe2602	2,6-dinitro-4-(trifluoromethyl)phenyl, Carbothioamide, Halophenyl	Unknown	Inhibits HRAS^G12V^ and *KRAS*^G12V^ mutation	[150]
†††††	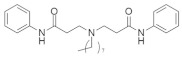	Phenylacetamide	Myeloid leukemia	Inhibits isoprenylcysteine carboxylmethyltransferase (ICMT)	[151]
Salirasib + FTS, Salirasib	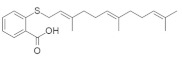 Salirasib 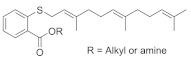 FTS, Salirasib	Trimethyldodecatrienylthiobenzoic acid	Bladder cancer (salirasib)	Inhibits glycolysis and oxidative phosphorylation pathways	[152] [153]

††††† = 3,3’-(ethylazanediyl)bis(*N*-phenylpropanamide).

**Table 6 ijms-23-03706-t006:** Miscellaneous small-molecule RAS inhibitors.

ID	Structure	Major Pharmacophore(s)	Cancer Type	Targeted Enzyme/ (M/A)	References
ML264	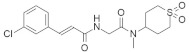	Dioxidotetrahydro-2*H*-thiopyran, Chlorophenyl	Colorectal cancer	KLF5 inhibitor	[160]
GDC-6036	Unpublished		Unpublished	*KRAS*^G12C^ inhibitor	Phase I Clinical trial (Genentech Inc)
LY3499446	Unpublished		Unpublished	*KRAS*^G12C^ inhibitor	Phase I and II Clinical trial (Eli Lilly)
D-1553	Unpublished		Unpublished	*KRAS*^G12C^ inhibitor	Phase I Clinical trial (InventisBio Co., Ltd., Shanghai, China)

## Data Availability

Not Applicable.

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
