# Peer review of "Small-Molecule RAS Inhibitors as Anticancer Agents: Discovery, Development, and Mechanistic Studies"

_ijms, 2022, doi:10.3390/ijms23073706_

Round 1

Reviewer 1 Report

Dear author

The theme choose for this review is one of great interest. Despite the effort in order to be published I have some recommandations. 

The abstract should be rewritten to be an abstract. Now it looks like an introductory part.
As a general remark, I would like a description of the compounds and structurally not only with code numbers. Figure 2 shows some functional groups / pharmacophores, which are not explained anywhere. The common structures and the way in which the mentioned structures are targeted should be highlighted.
As the title of the article refers to small molecules that aim ....., I consider that the biggest minus of this review results from the lack of discussions regarding structures / pharmacophores necessary for the affinity to the targets chosen from the signaling pathway.
This is also reflected in the part of the conclusions which is extremely vague. These also require that the abstract be rewritten.
I recommend that the authors strive to cover these shortcomings in order to turn the chosen topic into a high quality review.
I recommend the authors to publish this review after remedying the comments made, so with major revisions

Author Response

Comment: The theme choose for this review is one of great interest. Despite the effort in order to be
published I have some recommandations.
Response: Thank you for reviewing our manuscript. We have revised the manuscript following all the suggestions.
Comment: The abstract should be rewritten to be an abstract. Now it looks like an introductory part.
As a general remark, I would like a description of the compounds and structurally not only with code
numbers. Figure 2 shows some functional groups / pharmacophores, which are not explained anywhere.
The common structures and the way in which the mentioned structures are targeted should be
highlighted. As the title of the article refers to small molecules that aim ....., I consider that the biggest
minus of this review results from the lack of discussions regarding structures / pharmacophores
necessary for the affinity to the targets chosen from the signaling pathway.
Response: We have rewritten the abstract entirely. We have inserted an extra column highlighting the pharmacophores in Tables 1-6. Kindly note that the focus of this review is to discuss the mechanisms of action of the small molecules in the biological system. The qualitative and quantitive structure-activity relationship studies are not this review's theme.
Comments: This is also reflected in the part of the conclusions which is extremely vague. These also
require that the abstract be rewritten.
Response: We have completely rewritten the Abstract and Conclusions following the Reviewer’s
suggestion.
Comments: I recommend that the authors strive to cover these shortcomings in order to turn the chosen topic into a high quality review. I recommend the authors to publish this review after remedying the comments made, so with major revisions
Response: We have addressed the comments in our revised manuscript.

Reviewer 2 Report

In this review article, Shaila A. Shetu and Debasish Bandyopadhyay have summarized the most recent findings on the discovery and development of small molecule Ras inhibitors. For many decades, oncogenic Ras was considered as undruggable target molecule. The manuscript addresses the rapidly moving and very exciting field of Ras targeting that holds new promises in cancer therapy. The review is straightforward, clear, and well written. Therefore, I suggest the publication of the manuscript in the current form. 

Author Response

Comments: “In this review article, Shaila A. Shetu and Debasish Bandyopadhyay have summarized the most recent findings on the discovery and development of small molecule Ras inhibitors. For many decades, oncogenic Ras was considered as an undruggable target molecule. The manuscript addresses the rapidly moving and very exciting field of Ras targeting that holds new promises in cancer therapy. The review is straightforward, clear, and well written. Therefore, I suggest the publication of the manuscript in the current form.”
Response to the Reviewer:
Dear Reviewer,
Thank you so much for taking the time to evaluate our manuscript. A special thanks for your encouraging words.
Best regards,
Debasish Bandyopadhyay

Reviewer 3 Report

Shetu and Bandyopadhyay point in their nice article to a very important and issue. Development of small molecule RAS inhibitors is a brilliant performance of a forty years story of RAS research, given the dominating status of RAS oncogene in many cancers.

The manuscript is well written. There some aspect, which are needed to by revised:

  • Some small molecule inhibitors have additional names. For example, MRTX849 is called Adagrasib, and AMG 510 is called Lumakras or Sotorasib. It is of major importance to this article to include these names in the respective Tables as far as they exist.
  • The authors have clearly mentioned drug resistance in some case. Some very recent publications (e.g., doi: 10.1056/NEJMoa2105281) describe the development of heterogeneous responses to MRTX849 or AMG, bypassing their anti-growth effects by producing more of KRAS variants, to which these inhibitors do not bind. Specially the aspects of drug resistance and its future perspectives need to be discussed in more detail. The author should consider this issue in the section 6. Conclusions.
  • The notation of all types of RAS proteins (lowercase/uppercase) is quite mixed throughout the article, such as Ras, HRAS, H-Ras, Hras. The author may write them uniformly.
  • This article “doi: 10.1080/10409238.2018.1431605” needs to be referred to in the section 2.1 Ras Subfamily.
  • This article “doi: 10.3390/cells10071831” needs to be referred to in the section 2.2 Rho Subfamily.
  • Correct the Typo “Q16H” in Figure 2.
  • Residues Y96/H95/Q99 (line 368) are mutated in cancer cells in response G12Ci treatments (see doi: 10.1056/NEJMoa2105281).

Author Response

 Comment: Shetu and Bandyopadhyay point in their nice article to a very important and issue. Development of small molecule RAS inhibitors is a brilliant performance of a forty years story of RAS research, given the dominating status of RAS oncogene in many cancers.

The manuscript is well written. There some aspect, which are needed to by revised:
Some small molecule inhibitors have additional names. For example, MRTX849 is called Adagrasib, and AMG 510 is called Lumakras or Sotorasib. It is of major importance to this article to include these names in the respective Tables as far as they exist.

Response: We are thankful to the Reviewer for taking the time to evaluate our manuscript as well as for constructive suggestions. We have considered all the suggestions in our revised manuscript. We have included the names in appropriate places.

Comment: The authors have clearly mentioned drug resistance in some case. Some very recent publications (e.g., doi: 10.1056/NEJMoa2105281) describe the development of heterogeneous responses to MRTX849 or AMG, bypassing their anti-growth effects by producing more of KRAS variants, to which these inhibitors do not bind. Specially the aspects of drug resistance and its future perspectives need to be discussed in more detail. The author should consider this issue in the section 6. Conclusions.
Response: We thank to the Reviewer for this excellent suggestion. We have re-written the “Conclusion” accordingly.

Comment: The notation of all types of RAS proteins (lowercase/uppercase) is quite mixed throughout the article, such as Ras, HRAS, H-Ras, Hras. The author may write them uniformly.
Response: We have updated our revised manuscript following the suggestion. Moreover, we have written KRAS in italics when it indicates gene and in regular to indicate the protein. Thanks for drawing our attention to this matter.

Comment: This article “doi: 10.1080/10409238.2018.1431605” needs to be referred to in the section 2.1 Ras Subfamily.
Response: We have included this reference in the revised manuscript.

Comment: This article “doi: 10.3390/cells10071831” needs to be referred to in the section 2.2 Rho Subfamily.
Response: We have included this reference in the revised manuscript.

Comment: Correct the Typo “Q16H” in Figure 2.
Response: We have included done accordingly in the revised manuscript.

Comment: Residues Y96/H95/Q99 (line 368) are mutated in cancer cells in response G12Ci treatments (see doi: 10.1056/NEJMoa2105281).
Response: We have updated this information and included the reference in the revised manuscript.

Round 2

Reviewer 1 Report

Dear authors,
I appreciate that you have followed the made recommendations. At this time I consider that the review is suitable for publication.

Reviewer 3 Report

This nice article has now been significantly improved! I have no further comments.